# Upper Tropospheric Cloud Systems Derived from IR Sounders: Properties of Cirrus Anvils in the Tropics

Sofia E. Protopapadaki[1], Claudia J. Stubenrauch[1], and Artem G. Feofilov[1]

[1]Laboratoire de Météorologie Dynamique (LMD/IPSL), Sorbonne Universités,
UPMC Univ Paris 06, PSL Research, University, Ecole Normale Supérieure,
Université Paris-Saclay, Ecole Polytechnique, CNRS, Paris, France

*Correspondence to:* S. Protopapadaki (sprotopap@lmd.polytechnique.fr)

**Abstract.** Representing about 30% of the Earth's total cloud cover, upper tropospheric clouds play a crucial role in the climate system by modulating the Earth's energy budget and heat transport. When originating from convection, they often form organized systems. The high spectral resolution of the Atmospheric InfraRed Sounder (AIRS) allows reliable cirrus identification, both from day and night-time observations. Tropical upper tropospheric cloud systems have been analyzed by using a spatial composite technique on the retrieved cloud pressure of AIRS data. Cloud emissivity is used to distinguish convective core, cirrus, and thin cirrus anvil within these systems. A comparison with simultaneous precipitation data from the Advanced Microwave Scanning Radiometer - Earth Observing System(AMSR-E) shows that for tropical upper tropospheric clouds, a cloud emissivity close to 1 is strongly linked to a high rain rate, leading to a proxy to identify convective cores. Combining AIRS cloud data with this cloud system approach, using physical variables, provides a new opportunity to relate the properties of the anvils, including also the thinner cirrus, to the convective cores. It also distinguishes convective cloud systems from isolated cirrus systems. Deep convective cloud systems, covering 15% of the tropics, are further distinguished into single-core and multi-core systems. Though AIRS samples the tropics only twice per day, the evolution of longer living convective systems can be still statistically captured, and we were able to select relatively mature single-core convective systems by using the fraction of convective core area within the cloud systems as a proxy for maturity. For these systems, we have demonstrated that the physical properties of the anvils are related to convective depth, indicated by the minimum retrieved cloud temperature within the convective core. Our analyses show that the size of the systems does in general increase with convective depth, though for similar convective depth oceanic convective cloud systems are slightly larger than continental ones, in agreement with other observations. In addition, our data reveal for the first time that the fraction of thin cirrus over the total anvil area increases with the convective depth, similarly for oceanic and continental convective systems. This has implications for the radiative feedbacks of anvils on convection which will be more closely studied in the future.

## 1 Introduction

High clouds cover about 30% of the Earth (e.g Stubenrauch et al., 2013) and are of fundamental importance to climate as they modulate the Earth's energy budget and the heat transport in the upper troposphere, thus potentially influencing earth's

atmospheric circulation and water cycle. Their feedbacks still lack scientific understanding and heretofore represent a major uncertainty in predicting climate variability and change in climate models (Boucher et al., 2013).

In the tropics, where these high clouds are most abundant, they are part of large mesoscale systems of a characteristic size of tens of thousands $km^2$. They either form from organized deep convection or are directly formed *in situ* when cold air is supersaturated with water. This article focuses on the former in the tropics.

Within the last decade, numerous studies focused on these mesoscale convective cloud systems (MCS). Their structure and life cycle were studied by using composite techniques applied to satellite imagery and radar (e.g. Machado and Rossow, 1993; Machado et al., 1998; Del Genio and Kovari, 2002; Schumacher and Houze, 2003; Houze, 2004; Lin et al., 2006; Liu et al., 2007; Rossow et al., 2007; Yuan and Houze, 2010; Roca et al., 2014; Virts et al., 2015; Bouniol et al., 2016). These studies concentrated mainly on the thick cirrus anvils, because radar and visible-infrared imagery either miss or misidentify thin cirrus (Stubenrauch et al., 2013). However, the thinner cirrus are thought to be a part of the MCS's anvil that have a significant radiative impact which might regulate convection itself (Stephens et al., 2004; Lebsock et al., 2010). Their radiative forcing depends primarily on their horizontal extent, emissivity distribution, and the temperature difference with the underlying surface.

In addition, organized convection was studied by statistical analysis of cloud regimes defined by similar cloud pressure and optical depth within grid cells (Tselioudis and Rossow, 2011; Rossow et al., 2013; Stachnik et al., 2013; Tan et al., 2015; Oreopoulos et al., 2016). Though this approach proved to be very useful for advancing our knowledge on tropical convection, it does not provide information of the horizontal extent and structure of the systems. Recent studies which used the space-borne active instruments, lidar and radar, of the A-Train mission (Stephens et al., 2002) revealed the vertical structure of these systems (e.g. Luo et al., 2010; Yuan and Houze, 2010; Igel et al., 2014; Takahashi and Luo, 2014; Deng et al., 2016). They are, however, hampered by the very narrow track and thus are missing the horizontal extent of the system.

In this article we use infrared (IR) sounder data to study mesoscale deep convective systems, and more specifically their horizontal extent and IR emissivity distribution. The high spectral resolution of IR sounders, in particular the Atmospheric InfraRed Sounder (AIRS) aboard Aqua since 2002 and the Infrared Atmospheric Sounding Interferometers (IASI) aboard Metop since 2006, allow reliable cirrus identification, both from day and night time observations (e.g. Stubenrauch et al., 2010, 2013). Using the AIRS physical variables of pressure and emissivity, we reconstruct cloud systems. This approach distinguishes isolated cirrus from systems having convective core(s), using the IR emissivity as a proxy; an IR emissivity close to 1 has been verified to be closely related to larger rain rate and large-scale vertical ascent (2.2). This provides a unique opportunity to relate the properties of the anvils, including also the thinner cirrus, to those of the convective cores.

One of the World Climate Research Programme grand challenges is to determine the role of convection in cloud feedbacks (Bony et al., 2015). Compared to data bases of tropical mesoscale convective systems from radar and visible-infrared satellite imagery, this data base of upper tropospheric (UT) cloud systems from AIRS cloud properties includes cirrus, reliably identified down to an IR emissivity of 0.1 (corresponding to a visible optical depth of 0.2). The motivation of this article is to present this data base, which, coupled with other data, will provide observational metric for a better understanding of the interconnection between tropical convection and the heating induced by the outflowing anvils.

Proxies of convective intensity/strength or convective depth may be given by vertical updraft (e.g. Liu et al., 2007; Takahashi and Luo, 2014), lightning flash rate (e.g. Zipser et al., 2006), level of neutral buoyancy (e.g. Takahashi and Luo, 2014), area of heavy rainfall (e.g. Yuan and Houze, 2010), width of convective core (e.g. Igel et al., 2014), cold cloud top temperature or height (e.g. Machado and Rossow, 1993; Fiolleau and Roca, 2013) and mass flux (e.g. Tissier et al., 2016; Masunaga and Luo, 2016). While the level of neutral buoyancy describes the convective environment, the convective intensity is given by the strength of the vertical updraft, and the cloud top height can be considered as a proxy of convective depth. Therefore, these proxies might give insight into different aspects of convection. With AIRS alone, we are able to determine cloud top height temperature and, therefore, to explore the anvil properties in relation to the convective depth.

Details on the AIRS cloud retrieval and the construction of UT cloud systems are given in section 2. Results on the statistical properties of tropical upper tropospheric cloud systems, including maturity stage with respect to the convective core fraction, are given in section 3. Relationships between convective depth and anvil properties of tropical mature convective cloud systems are then discussed in section 4. Conclusions and an outlook are presented in section 5.

## 2   Methodology

### 2.1   Cloud properties derived from AIRS observations

AIRS is a high spectral resolution infrared spectrometer, aboard the polar orbiting EOS Aqua satellite with an equatorial crossing at 1:30 a.m. and 1:30 p.m. local time (Chahine et al., 2006). AIRS completes approximately 14.5 orbits per day with each orbit "swath" being of $48.95°$, divided by 90 footprints for each scan line. The spatial resolution of a footprint varies from about 13.5 km x 21 km at nadir to 41 km x 21 km at the scan extremes.

The LMD cloud property retrieval is based on a weighted $\chi^2$ method and uses eight spectral channels sounding along the 15 $\mu m$ $CO_2$ absorption band (Stubenrauch et al., 2010), providing cloud pressure $p_{cld}$ and emissivity $\epsilon_{cld}$ of a single cloud layer (of the uppermost cloud layer in the case of multi-layer clouds). By introducing empirical weights, the method takes into account the vertical weighting of the different channels, the growing uncertainty in the computation of $\epsilon_{cld}$ with increasing pressure and uncertainties in atmospheric profiles (Stubenrauch et al., 1999). A crucial consideration in the cloud retrieval is the determination of clear sky and opaque cloud radiances $I_{clr}$ and $I_{cld}$, since $\epsilon_{cld}$ is defined as $\epsilon_{cld} = (I_{meas} - I_{clr})/(I_{cld}(p_{cld}) - I_{clr})$. For their computation we need temperature profiles and surface skin temperature as well as atmospheric transmissivity profiles at the corresponding wavelengths for the atmospheric situation of the measurements. The atmospheric spectral transmissivity profiles have been simulated by the 4A radiative transfer model (Scott and Chédin, 1981; operational version available at http://www.noveltis.net/4AOP), separately for each satellite viewing zenith angle and for about 2000 representative clear sky atmospheric temperature and humidity profiles of the Thermodynamic Initial Guess Retrieval (TIGR) data base (Chevallier et al., 1998; Chédin et al., 2003). Since IR sounders, in combination with microwave sounders, were originally designed for the retrieval of atmospheric temperature and humidity profiles, the atmospheric clear sky situation can then be directly described by simultaneously retrieved AIRS L2 atmospheric profiles  (Susskind et al., 2014) of good quality (when the situation is not too cloudy), provided by NASA (Version 6 available at Goddard Earth Sciences

Data and Information Services Center). In the other case, the instantaneous profiles and surface skin temperature are replaced by those of good quality, averaged over 1°latitude x 1°longitude, or interpolated in time. The proximity recognition between these AIRS L2 atmospheric profiles and the TIGR atmospheric profiles is described in detail in Stubenrauch et al. (2008). Once $p_{cld}$ and $\epsilon_{cld}$ are retrieved by the $\chi^2$ method, cloud temperature $T_{cld}$ is determined from $p_{cld}$, by using the AIRS temperature profile.

Recently, we have developed a modular cloud retrieval code (CIRS, Clouds from IR Sounders, Feofilov and Stubenrauch, 2017), which can be applied to any IR sounder data. To derive a 13-year global climatology of cloud properties from AIRS (2003-2015), we used the latest ancillary data (atmospheric profiles, surface emissivities and atmospheric transmissivities). Compared to the version which is distributed at the French data center ICARE and which has been evaluated in the Global Energy and Water Exchanges (GEWEX) cloud assessment (Stubenrauch et al., 2013), these cloud data are very similar for high-level clouds and slightly improved for low-level clouds (Stubenrauch et al., 2017). Cloud types can then be defined according to $p_{cld}$ and $\epsilon_{cld}$.

In this analysis, to facilitate the reconstruction of the UT cloud systems from the AIRS cloud properties, it is convenient to grid the data, keeping the statistics and occurrence of the individual cloud types inside the grid, while the physical parameters (T, p, $\epsilon$) are averaged inside the grid cell. The grid cell size should not be greater than the average size of the smallest cloud system. A good compromise was found by introducing grid cells of 0.5° in latitude and longitude.

## 2.2 Construction of upper tropospheric cloud systems

Before reconstructing the horizontal extent of the UT cloud systems, a critical question has to be addressed: how to define UT clouds? Most studies on tropical MCS's life cycle and structure used the IR brightness temperature ($T_B$) to define cold clouds: Yuan and Houze (2010) merged adjacent footprints containing cold clouds defined as those with $T_B$ <260 K, while Machado et al. (1998) used $T_B < 245$ K and, Fiolleau and Roca (2013) and Roca et al. (2014) have considered only footprints with $T_B$ <233 K. However, $T_B$ depends both on cloud altitude and opacity; opaque clouds ($\epsilon \approx 1$) have an IR brightness temperature close to the cloud top temperature $T_{cld}$, though it can happen to be a few degrees lower (Sherwood et al., 2004; Stubenrauch et al., 2010). For optically thinner clouds the radiation reaching the satellite instrument includes, in addition to the cloud's emission, also a fraction of the emission from the warmer earth's surface and the atmosphere passing through these semi-transparent clouds. Figure 1 presents the median value and the quartiles of $T_B$ as well as $T_{cld}$ as a function of $\epsilon_{cld}$ for high clouds in the tropics, with $p_{cld} < 440$ hPa (corresponding to a height of about 7 km), a definition generally found in the literature (e.g. Rossow et al., 1999; Stubenrauch et al., 2012). One observes that indeed $T_B$ increases with decreasing $\epsilon_{cld}$ while $T_{cld}$ does not show a particular relationship, except that the most opaque clouds in the tropics seem to be on average also colder. The relation between $T_B$, $T_{cld}$ and $\epsilon_{cld}$ is further explored in Fig. 2; $T_B$ corresponds to $T_{cld}$ only for opaque clouds, whereas for a given $T_B$ the associated cloud might also be optically thin and colder. This means that for instance, an upper threshold of 245 K on $T_B$ will include higher clouds ($T_{cld} < 245$ K approx. 8 km) with an emissivity down to 0.7, but their cloud temperature might then be underestimated by several 10's of °C. This only allows relating the thickest anvil properties to convection.

The present cloud system approach, employing cloud altitude (temperature, pressure) and opacity (emissivity) has the advantage of a clear distinction between high and low clouds based on cloud pressure, and of thin and thick cirrus, based on cloud emissivity. This is important, since, as discussed in the introduction, this new UT cloud system approach aims to explore the horizontal structure of the UT cloud systems, including thin cirrus.

Since the AIRS initial spatial resolution is more adapted to study organized convection rather than small scale shallow convection, we revise the definition of upper tropospheric clouds i) towards slightly higher clouds and ii) by using a tropopause dependent definition. Hereafter, UT clouds will be considered as those being at most 250 hPa below the tropopause corresponding to a maximum cloud pressure of about 350 hPa and a height of about 8 km in the tropics. It should be stressed that the standard high cloud definition of $p_{cld}$< 440 hPa high has also been tested and the results are coherent with those reported in sections 3 and 4.

Typically, a convective system is composed of an opaque precipitating core which detrains cirrus in the form of an anvil at the height of neutral buoyancy (e.g. Luo et al., 2010; Takahashi and Luo, 2014). To investigate whether cloud emissivity can be used as a proxy to identify convective cores, AIRS cloud data have been collocated with simultaneous AMSR-E precipitation data (Kummerow and Ferraro, 2006). Figure 3 presents median values and quartiles of the maximum and average rain rate from the AMSR-E measurements (of spatial resolution of about 5 km at nadir) and the vertical wind at 500 hPa from the meteorological reanalysis ERA Interim (Dee et al. , 2011), within a grid cell, as a function of cloud emissivity averaged from AIRS UT clouds within the same grid cell. The vertical wind data are interpolated, spatially from 0.75° to 0.5° grid cells, and temporally from 6 hourly universal time to 1:30 AM and PM local time. A strong positive correlation between cloud emissivity and precipitation is observed for high $\epsilon_{cld}$. The rain rate might be of stratiform origin even when $\epsilon_{cld}$ is close to 1, as according to the quartile bands there is a probability that the average rain in the 0.5° grid is below 1 mm h[-1], while the maximum RR quartile is at 6 mm h[-1]. The link with convection can only be shown through vertical updraft. Though the ERA Interim vertical wind has a low horizontal spatial resolution (about 0.75°) and is therefore quite diluted, Figure 3 shows that opaque clouds are in general linked to stronger large-scale vertical ascent. Based on figure 3, hereafter, convective cores are defined as those with $\epsilon_{cld}$>0.98. The emissivity threshold for distinguishing cirrus from thin cirrus was set to 0.5 as i) below this threshold no rain occurs at all (not shown), ii) this threshold has already been used to define thin cirrus in earlier IR sounder analyses, iii) the studies exploring tropical convective cloud systems using IR brightness temperatures exclude all high clouds with an emissivity below this value (Fig. 2).

To study the horizontal extent of cloud systems, a full spatial coverage is required. However, in the tropical region where the cloud systems will be explored, $30°N - 30°S$, AIRS measurements only cover about 70% of the surface due to gaps between orbits (e.g Fig.1 of Feofilov et al., 2015). Thus, the missing data have to be extrapolated from the properties of the cloud types determined around the gaps. It should be stressed that days with missing orbits are completely excluded from the analysis; only scenes with coverage above 68%, representing more than 85% of the total statistics, are considered. In the following we describe the method developed to fill the missing data gaps. In each grid cell of 0.5°x0.5° the distribution of the number of measurements per cloud type is known. Cloud type distributions in empty grid cells are obtained from the probability density function (PDF) of the neighbouring grid cells. The PDF of an empty grid cell is built as the sum of the neighbouring PDFs,

normalized to 1, weighted by the inverse squared root of the distance between the grid cells. Similarly, the physical properties of each cloud type in the interpolated grid, such as temperature, pressure and emissivity, are computed using the same weighted average method.

In the course of the study several questions emerged, such as how many neighbours to use, and what should be the shape of the region for the neighbours to be included in the interpolation. The reason we draw readers' attention to these details is due to the irregular gap area shape and size which varies with latitude. The optimal filling configuration was deduced by statistically comparing the fractions of each of the UT cloud types in the grid cells with real data and those with interpolated data, but also by visually examining geographical maps of cloud types such as the top panel of Figure 4. We found that the most appropriate way to get an UT cloud amount in the gaps consistent with the one in the data grid cells, while preserving cloud system shapes, was to choose a number of neighbours proportional to the distance between the grid cells-to-be-filled and the closest non-empty-grid cell. By doing so, an empty grid cell surrounded by non-empty grid cells is filled using only a small number of the proximity data neighbours, while a cell located at the center of a gap near the equator (gap with maximum horizontal width reaching 700 km) is filled using a larger number of data (up to 100 grid cells) since the uncertainty is higher. The filling algorithm first scans eastward and westward of a grid cell to-be-filled to count the number of empty grid cells in both directions until a non-empty grid cell is found; the closest distance being the gap reference distance. Then, a spiral scan over the neighbours is performed for a number of cycles which increases linearly with the gap distance. From case studies we observed that obtaining realistic cloud system shapes requires the scan to be bound vertically to $\pm 3$ grid cells, while allowing the horizontal scan free. As an example, the top panel in Figure 4 presents a geographic map of cloud types for one day in July after the data gaps filling.

Once the gaps are filled, we apply a composite technique to reconstruct the upper tropospheric cloud systems; adjacent grid cells containing UT clouds and sharing a common side are grouped. The grid cells must contain more than 70% of UT cloud types within all AIRS measurements in order to be considered in the procedure. For interpolated grid cells the threshold is set slightly lower, to 65%, as this 5% difference corrects for an observed bias in the UT cloud amount of the interpolated areas. To ensure the spatial continuity of cloud systems, the average cloud pressure difference between two adjacent grid cells must be lower than 50 hPa; this is a legitimate value as it is slightly above the uncertainty of retrieved $p_{cld}$, which is of the order of 30 - 40 hPa (Stubenrauch et al., 2012; Feofilov and Stubenrauch, 2017).

To identify opaque areas inside the built UT cloud systems, which potentially enclose convective core(s), a second grouping is performed. The emissivity limit for the opaque area definition is set to 0.9. The cloud system is then considered as a "convective" one when containing at least one grid cell with $\epsilon_{cld} > 0.98$ within the opaque area. The above core identification procedure provides the number of convective cores in a cloud system and thus allows its classification as non-convective, if no convective core is found, or as convective if at least one core is found. The latter are further classified, with respect to the number of cores, to single-core and multi-core systems.

Figure 4 bottom panel present for the same day as in the top panel, the tropical cloud systems including opaque and convective core areas. Middle panel presents a more detailed horizontal structure of the UT cloud systems, illustrating the anvil amount not taken into account in the various analyses which use IR $T_B$ mentioned in section 2.2.

# 3 Exploration of Tropical Upper Tropospheric cloud systems

## 3.1 Statistical properties

We find that upper tropospheric cloud systems cover about 20% (25%) of the tropical band, defined as 30° N-30° S (15° N-15° S). Their horizontal extent varies significantly, starting from a single grid cell with a size of about 2500 km$^2$, reaching several 10$^8$ km$^2$. These UT cloud systems may be distinguished as convective or non-convective (isolated cirrus) systems. More specifically, convective (single and multi-core) systems cover 15% (20%) while isolated cirrus systems cover 5% (5%) of the tropical band 30° N-30° S (15° N-15° S). The latter might originate from convection or formed by *in situ* freezing. Studies using Lagrangian transport performed by Luo et al. (2004) and Riihimaki et al. (2012) have shown that about 50% of these isolated cirrus systems form *in situ* while the other half corresponds to dissipating convective systems. Table 1 summarizes the statistical repartition of tropical isolated cirrus systems, single-core and multi-core convective systems in the 30° N-30° S band along with their average sizes. Though isolated cirrus systems significantly outnumber the convective systems, their average horizontal extent is a factor of 10 smaller than the one of single-core convective systems. Multi-core convective systems are significantly larger than the other categories, compared to single-core by a factor of 20, while representing only 1% of the population. Among convective cloud systems, those having horizontal extent larger than 3*10$^8$ km$^2$ represent about 10% and are mainly located over the western Pacific during the monsoon period (Liu et al., 2007); a region with warm surface temperatures, large convective mass fluxes (Tissier et al., 2016), and large UT humidity (Virts et al., 2015; Houze et al., 2016). This region is also known for building mesoscale convective complexes (e.g. Mapes and Houze, 1993; Deng et al., 2016), including several convective systems, often in different phases of development and connected by ubiquitous thin cirrus.

Figure 5 presents geographical maps of occurrence a) isolated cirrus and b) all convective cores, together for single and multi-core systems, also separately for c) boreal winter and d) boreal summer, and e) of single core convective systems. The convective activity pattern clearly follows the Intertropical Convergence Zone (ITCZ) with maxima observed over the warm pool, north west South America and central Africa, and over the summer hemisphere. The patterns are in agreement with previous findings obtained from the International Satellite Cloud Climatology Project (Tan et al., 2015), the CloudSat mission (Igel et al., 2014), the Tropical Rainfall Measuring Mission (Houze et al., 2015) and from geostationary satellites Fiolleau and Roca (2013). As expected, isolated cirrus are abundant and are found to cover wide areas in the vicinity of the convective active regions.

## 3.2 System composition and life-cycle stages

As discussed in the introduction, the impact of UT cloud systems on the Earth's energy budget depends on their horizontal extent, their emissivity distribution, and the temperature difference between the cloud and its underlying surface (lower clouds or earth surface). The latter has been explored by Haladay and Stephens (2009). In this work, the first two points will be studied.

Hereafter, we are primarily interested in the horizontal cloud system emissivity structure, rather than in the total coverage over the tropical band, and to keep the uncertainties low, we consider only convective cloud systems which are composed of more than 80% of real data.

Figure 6 presents the average proportion of convective core, thick and thin anvils as a function of the UT cloud system horizontal extent, separately for single and multi-core convective systems. The statistics include convective systems at different phases of their life cycle (in development and mature). As the systems get larger, the fraction of the convective core decreases to 10% and that of thin cirrus anvil increases up to about 30%. The same tendencies are observed for both single and multi-core systems, with the only difference that the latter have slightly smaller fractions of convective core area and slightly larger fractions of thin cirrus area.

The composition of a convective system (convective part, thick and thin anvil) depends on the system life-cycle stage (as illustrated in Fig. 9d of Machado et al. (1998)). Our analysis, using snapshots which are available only every twelve hours, cannot directly track the life cycles of the convective systems. However, in particular organized convection often has life time longer than 24 hours; it has already been demonstrated in previous studies using satellite data with better temporal resolution (Machado et al., 1998 and Futyan and Del Genio, 2007) or with varying observation time (Fiolleau and Roca, 2013) that the largest systems have the longest life cycle, up to several days. Therefore, even with only two measurements per day we should be able to observe systems in different phases of their life cycle and explore them statistically. Our article is not focused on studying the life cycle itself, but aims to select relatively mature convective systems, for which one can then explore the relationship between anvil properties and convective depth. We use as proxy of maturity stage the fraction of convective core horizontal extent with respect to the total cloud system horizontal extent. This variable has been proven to be an indicator of convective cloud maturity as it follows the life cycle in high temporal resolution studies using IR imagery of geostationary satellites (Machado et al., 1998), Tropical Rainfall Measuring Mission (TRMM) (Fiolleau and Roca, 2013), as well as using CloudSat radar (Bacmeister and Stephens, 2011), the latter taking data at the same observation time as AIRS.

Figure 7 shows the normalized distribution of convective core fraction, separately for single and multi-core systems. In general, this fraction has a wider distribution and peaks at a larger value for single-core systems compared to multi-core convective systems (at 0.25 and 0.1, respectively). A small fraction of single-core convective systems consists only of the convective core itself; these are systems in the development phase. Only during maturity and dissipation do convective systems include increasing upper tropospheric stratiform cirrus anvils, while the fraction of the convective area decreases (e.g Leary and Houze, 1979; Machado and Rossow, 1993). Multi-core convective systems, agglomerating systems probably in different stages of development, are not suitable for exploring the system's life cycle, and therefore will not be considered in this study. Moreover, in order to ensure a purified sample of single-core systems we exclude single-core systems which have more than one opaque area ($0.9 < \epsilon_{cld} < 0.98$).

By stratifying single-core convective systems according to their fraction of convective area within the cloud system, we explore whether their physical properties follow an evolution pattern which corresponds to different life cycle stages. To do so, taking in account the convective fraction distribution of single core systems of Fig 7, we consider eleven intervals of equal statistics with respect to the convective fraction: [1, 0.78, 0.65, 0.55, 0.47, 0.40, 0.34, 0.29, 0.24, 0.19, 0.13, 0.01], indicated as 11 "maturity steps" in Figures 8 and 9.

Single core systems over land and ocean, the former having a fraction of land convective grid cells over total convective grid cells above 0.5 and the latter below 0.5, are further separated to early afternoon (PM) and night (AM), since diurnal variations

are expected. The statistics at each "maturity step" is shown in Fig. 8. One observes slightly more "developing" systems over land and more dissipating systems over ocean in the early afternoon. During night the statistics are more equally distributed, with twice as many oceanic single-core convective systems than in the afternoon. These findings are in agreement with studies on tropical precipitation which show a peak in the late afternoon over land and a few hours before sunrise over ocean (e.g. Liu and Zipser, 2008). One has to keep in mind that our specific observation times might not capture the peak of convection.

Figure 9 presents the median values of the physical properties of single-core convective systems for successive life cycle stages, separately in the early afternoon and at night, over land and over ocean. The total cloud system horizontal extent (Fig. 9a) increases during the whole life cycle, something expected as the detrained anvil increases as the system gets older. We do not capture the anvil shrinking as shown in Fig. 9a of Machado et al. (1998), most likely, because Machado et al. studied only the thicker anvils when using the IR $T_B$ (Fig. 2) and the thinner anvil part increases towards dissipation (Fig. 9c). Moreover, our convective system definition requires at least one convective grid cell and therefore the system is not captured in its advanced dissipation. The horizontal extent of the convective core (Fig. 9b) increases until it reaches a plateau around life cycle stage 5-9, which corresponds to a convective fraction between about 0.1 and 0.3. The behavior is similar over land and ocean, except for ocean in the early afternoon, where the increase in convective core size is stronger with a peak for cloud systems with a convective fractional area of about 0.2. When considering the evolution of the emissivity distribution within the convective system (Fig. 9c) and the ratio of thin cirrus over cirrus within the anvil (Fig. 9d), the average emissivity of the cloud system decreases and moreover the fraction of thin anvil increases along the system's life cycle in agreement with expectations. It is interesting to note that the behavior is similar over ocean and over land. Rain rate is maximum at the developing phase and decreases successively until dissipation (Fig. 9e), with twice higher rates over land than over ocean. This finding is in agreement with Fig. 5 of Fiolleau and Roca (2013).

The minimum temperature of the convective core is the only variable which does not have a clear behavior. When considering specific regions, like the three land regions and three ocean regions discussed in (Liu and Zipser, 2008), the behavior is similar as in Figure 9, with similar $T_{min}^{cb}$ over all maturity steps in the Atlantic and East Pacific, while for the other regions $T_{min}^{cb}$ slightly increases with decreasing convective fraction (see Fig. 1e of the supplement). However, all minimum temperatures of the convective cores seem to converge towards a plateau for the mature and dissipating convective systems. It is interesting to note that the regional spread of $T_{min}^{cb}$ and of average convective core rain rate, the latter again larger over land than over ocean regions, both variables indicating convective strength, is larger in the developing stage, whereas the regional spread of variables linked to the areal properties of the systems like system and convective core size is larger towards the mature and dissipating stage. Regional spread of average emissivity and ratio of thin cirrus over total anvil have relatively small and constant spread during the whole development, of about 0.05 and 0.1, respectively (Figs. 1c and 1d of supplement). These regional analyses confirm that the physical properties of convective systems have a similar behavior (except $T_{min}^{cb}$), when using the convective fraction as a proxy for maturity, with regional spreads probably due to the different environmental features affecting these regions.

We are interested to study the relationships between anvil properties and convection when the systems are mature. Therefore we are confident to isolate these systems according to Fig. 9b) by requiring a convective fraction within the system between 10 and 30%, leading to averages in thin cirrus over cirrus anvil of about 30%.

## 4    Relationships between convective depth and cirrus anvil properties

5    As discussed in the introduction, there are different proxies describing the convective intensity/strength or convective depth, which might give an insight into different aspects of convection. The level of neutral buoyancy (LNB), which can be computed from atmospheric soundings, describes the convective environment and sets the potential vertical extent for convective development (Takahashi and Luo, 2014). In general, convective intensity is given by the strength of the vertical updraft. A strong updraft should produce a large radar echo top height (ETH) and therefore a smaller difference between cloud top height (CTH) and ETH, i.e. large particles are lofted to greater altitude. Using CloudSat radar data, Takahashi and Luo (2014) have shown that for deep convective systems CTH correlates well with LNB. CTH and ETH are also positively correlated, suggesting that convective intensity and convective depth are related. Unlike the relationship between CTH and LNB, larger correlations between CTH and ETH are found over land than over ocean. CTH is linked to cloud top temperature $T_{cld}$ through the atmospheric temperature profile. Therefore, with AIRS data alone, we are able to explore the anvil properties with respect to CTH by using $T_{cld}$ of the convective part of the cloud systems.

In the following, we will investigate convective depth only for mature convective systems, defined according to Fig. 9 as systems for which the fraction of convective core area varies between 0.1 and 0.3. It should be stressed that all the correlations obtained in this section are very well reproduced if a tighter convective fraction interval is used as maturity proxy. The convective depth of a mature cloud system can be deduced by its height, and therefore by the cloud top temperature of the convective core; the latter being directly linked to the height through the atmospheric temperature profile. Since the convective core might also include parts of the rainy anvil, we use as a proxy for convective depth the minimum temperature within the convective core ($T_{min}^{cb}$) instead of the average $T_{cld}$ of the convective core. Fig. 10 presents the relation between $T_{min}^{cb}$ and the maximum rain rate within the convective core (at a spatial resolution of 5 km), separately over land and ocean for mature single-core systems. By using the maximum rain rate at such a spatial resolution the probability should be higher to correspond to convective rain. From Fig. 10 we deduce that the colder (higher) the convective core, the higher is the maximum rain rate. The relationship is similar over ocean and over land for $T_{min}^{cb}$ values larger than about 210 K, whereas for cloud systems with the lowest $T_{min}^{cb}$ values the maximum rain rate inside the convective core gets significantly higher over land. This is in agreement with earlier findings of Schumacher and Houze (2003) and Liu et al. (2007).

Figure 11 top panel, presents the size of the mature convective systems as a function of the minimum temperature within the convective core, separately for oceanic and for continental systems. We observe an increase of the size of the systems with increasing convective depth, represented by decreasing $T_{min}^{cb}$. Similar results are obtained from regional analyses (Fig 2a of supplement), indicating the robustness of this finding as in all regions the size of the systems increase with decreasing $T_{min}^{cb}$.

It should be however stressed that while the slopes for Atlantic and East Pacific compare to the oceanic slope in general, the slope of the West Pacific is similar to the one of land regions.

Whereas it is straightforward to determine the minimum temperature within a single-core convective system, it is more difficult to consider this proxy for multi-core convective systems. The latter might be composed of several convective sub-systems in different phases of development. Nevertheless, we build for those systems the average $T_{min}^{cb}$ over all convective cores of the system. Considering the bottom panel of Fig. 11, one observes that multi-core convective systems behavior is analogous to single-core systems.

From Fig. 10 and 11, we conclude that for both single and multi-core systems, oceanic convective systems of a similar convective depth as continental systems have a larger size with less intense convective rain, a behavior significantly enhanced for systems with an important convective depth. This difference in structure was already pointed out in earlier studies (e.g. Liu et al., 2007). Furthermore, Liu et al. (2007) have shown that tropical continental mesoscale convective cloud systems are in general smaller in size than oceanic systems, though the vertical updraft and horizontal extent of the convective cores are in general larger, while their convective depth is similar. Their ice water path is also larger than the one of oceanic systems, which is caused by different microphysics between land and ocean  (Sohn et al., 2015). An exception seems to be the West Pacific which seems to behave like land regions (Fig. 2a of supplement). Findings by Zipser et al. (2006) and Hamada et al. (2015) suggest that extreme convective events correspond to cloud systems with a smaller size than those of extreme rainfall events. This demonstrates that different variables give insight into different aspects of convection. One has also to keep in mind that the development of a convective system into maturity spans a certain time interval: one expects that a large updraft leads to a convective system of large height (low $T_{min}^{cb}$) which then develops horizontally. Aerosols and humidity also play a role in invigorating convection (e.g. Altaratz et al., 2014). It has been shown that the rain rate decreases during the development and most probably also the strength of the vertical updraft. This means that one needs to undergo time-lag studies or Lagrangian transport studies which follow closely the convective systems and their atmospheric environment to advance further. While these detailed studies will be subject of a follow-on study, which will include additional variables from other data sets, such as ERA5, we can still go one step further with AIRS data alone, by investigating the next question: will there also be a difference in the anvil horizontal structure with increasing convective depth?

Figure 12 top panel shows the ratio of thin cirrus anvil area over total anvil size as a function of $T_{min}^{cb}$ for mature single-core convective systems, separately over land and over ocean. With increasing convective depth (decreasing $T_{min}^{cb}$) the fraction of thin cirrus anvil increases, and this in the same manner for oceanic convective systems as for continental systems. Regional analyses confirm the correlation between the fraction of thin cirrus over total anvil and the convective depth (Fig. 2b of supplement) with similar slopes, though the absolute values show a spread of about 0.1. Systems over the East Pacific and the Atlantic are slightly more opaque, while anvils over Indonesia are those containing a larger fraction of thin cirrus. Bottom panel in Fig. 12 presents the same quantities, for land and oceanic systems together, separately for single and multi-core systems. We observe a very similar behavior for single-core convective systems and multi-core systems. These results indicate a robust relationship between convective depth and the horizontal emissivity structure of the detrained cirrus anvils. To go one step further in our investigation, Fig. 13 shows how the cirrus anvil emissivity varies with increasing distance to the

convective core, normalized by dividing with the square root of the size, for mature single-core systems. Three intervals of $T_{min}^{cb}$ are considered, representing systems of different convective depth. For all systems the cirrus anvil emissivity decreases with increasing distance, as one would expect. While the decrease in emissivity is comparable for all systems within the first quarter of the horizontal extent, it continues to decrease more rapidly for systems with strong convective depth compared to those reaching lower altitudes. This might have important implications for the radiative impact of these systems in relation to increasing convective intensity in a warming climate (Tan et al., 2015; Bony et al., 2016).

## 5 Conclusions and Outlook

We have built Upper Tropospheric cloud systems, using cloud pressure and emissivity retrieved from 13 years of AIRS observations. These data have been used to investigate the properties of tropical UT cloud systems, and in particular relationships between the convective depth, given by cloud top temperature in the mature stage of a convective cloud system, and the properties of the surrounding cirrus anvils.

The benefits of the present UT cloud system data base compared to other data and methods are that 1) IR sounder data have a large instantaneous coverage and are sensitive to thin cirrus down to an emissivity of 0.1 (0.2 in visible optical depth), day and night, and 2) our cloud retrieval provides the physical properties of altitude and emissivity decoupled, allowing to reconstruct the horizontal extent of the UT cloud systems and then to distinguish between deep convective cloud systems and isolated systems and to resolve their emissivity structure, essential to determine the radiative feedback of the anvils on convection. For our investigation we first needed to establish proxies to identify 1) convective cores, 2) mature deep convective systems and 3) the convective depth of a mature convective system. It was demonstrated, using rain rate and large-scale vertical winds, that in the tropics UT opaque clouds with an emissivity close to 1, have a large probability to stem from convection, even though they include probably a part of stratiform rain. Therefore the cloud emissivity permits to differentiate convective cores, cirrus and thin cirrus anvils as well as to identify single-core and multi-core convective systems. UT cloud systems cover about 20%-25% of the tropics. While the frequency strongly decreases from isolated cirrus towards multi-core convective systems, the latter's coverage is the largest. The fractional area of the convective core within a cloud system has already been proven to be a maturity stage proxy. Though considering only two measurements per day, the evolution of properties of single-core convective systems could still be statistically followed by using convective fraction within a cloud system as a proxy for maturity, since our results are compatible with findings using a better temporal resolution. The size of the convective core reaches a plateau and then decreases during the stage of dissipation, guiding us to define mature convective systems as those with a convective core fraction between 0.1 and 0.3.

Several proxies of convective intensity/strength or depth exist, giving insight into different aspects of convection. With our data, we could probe mature convective cloud system's characteristics with respect to the convective core minimum temperature, a variable indicative of the convective depth. It could be shown that colder convective systems (meaning those rising higher) have larger values of maximum rain rate within the convective core, a tendency more marked over land, as well as larger cirrus anvils, a tendency more marked over ocean. Both findings are in agreement with previous studies. Compared to

other methods, our approach provides the unique opportunity to study also the horizontal emissivity structure within the anvils. It was revealed that the fraction of thin cirrus over the total anvil area increases with increasing convective depth, similarly for oceanic and continental mature convective systems and both for single and multi-core systems. Regional analyses, besides some observed amplitude variations over the same surface type, confirmed these tendencies. We also demonstrated that with increasing convective depth, the emissivity of the anvil decreases in general more sharply with increasing distance to the convective core. This might have important implications for the radiative effects of these systems, in relation to a convection intensity increase in a warming climate.

The above findings are very promising and the observed relationships might provide observational metrics for studying detrainment processes with Cloud Resolving models or even climate models, if their spatial resolution is similar to the one of our data base, and for constraining parameterizations related to convection and detrainment. Combined with variables derived from other data sets, such as vertical cloud structure and corresponding heating rates, atmospheric humidity, surface temperature, level of neutral buoyancy, vertical and horizontal winds, his data bas will be the basis to address questions on feedbacks between anvils and convection and on their modulation of the atmospheric circulation, in particular in respect to climate change. Furthermore, Lagrangian transport analysis could be used to indicate the origin of the isolated cirrus systems and to assess the link between convective sources and the air entering the stratosphere. Moreover, when meteorological reanalyses are available at higher spatial and temporal resolution, exploration of lag correlations between variables such as vertical winds, size of convective core, rain rate, and other atmospheric condition variables, could give a better understanding of convection mechanisms.

*Acknowledgements.* This research was supported by the Centre National d'Etudes Spatiales (CNES), the Centre National de la Recherche Scientifique (CNRS) and the European Space Agency (ESA), within its framework of ESA Climate Change Initiative. The authors thank the Earth Observing System AIRS and AMSR-E teams for providing the data. The calculations have been performed at the ClimServ IPSL centre. The authors also thank two anonymous reviewers who helped to improve this manuscript through their thoughtful comments.

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

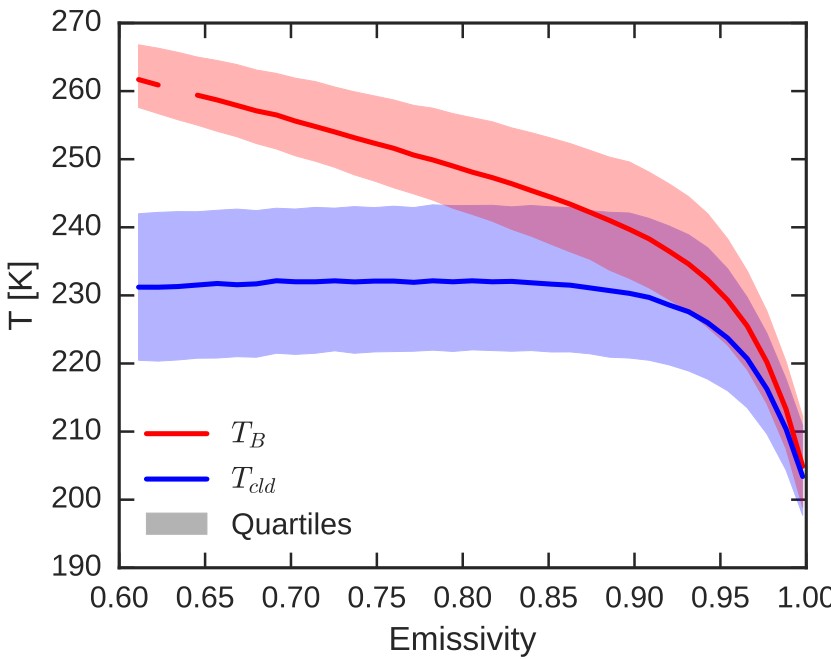

**Figure 1.** Median and quartiles of cloud IR brightness temperature (red) and retrieved cloud temperature (blue) as a function of cloud emissivity for high clouds ($p_{cld}$<440 hPa) identified from AIRS observations in the tropics, at a spatial resolution of 0.5°. Statistics for January and July 2006-2007.

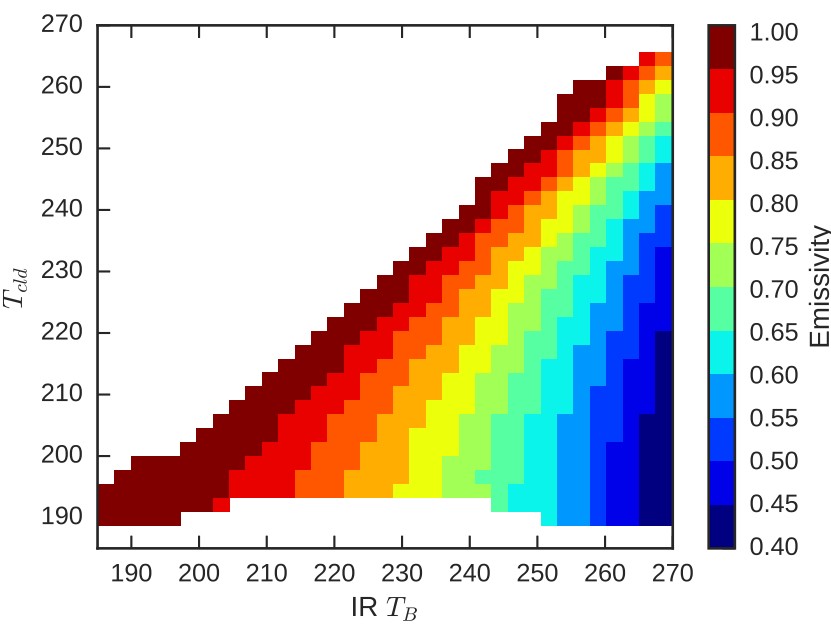

**Figure 2.** Average value of cloud emissivity for bins of $T_{cld}$ and $T_B$ for high clouds ($p_{cld}$<440 hPa) identified from AIRS observations in the tropics, at a spatial resolution of $0.5°$. Statistics for January and July 2006-2007.

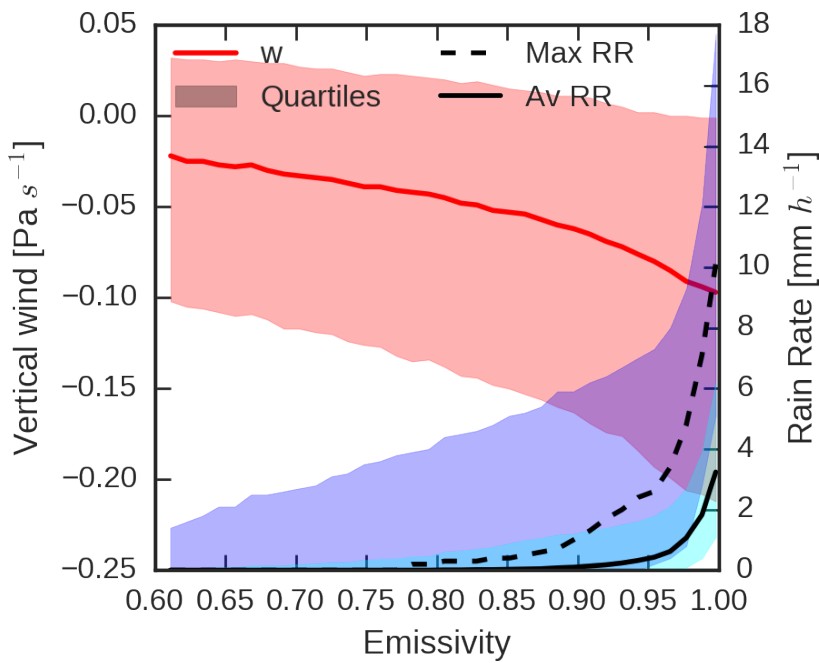

**Figure 3.** Median and quartiles of maximum (dashed black) and average (solid black) rain rate from AMSR-E, and average vertical winds (solid red) from ERAI, as a function of cloud emissivity for high clouds ($p_{cld}$<440 hPa) identified from AIRS observations in the tropics, at a spatial resolution of $0.5°$. Statistics for January and July 2006-2007.

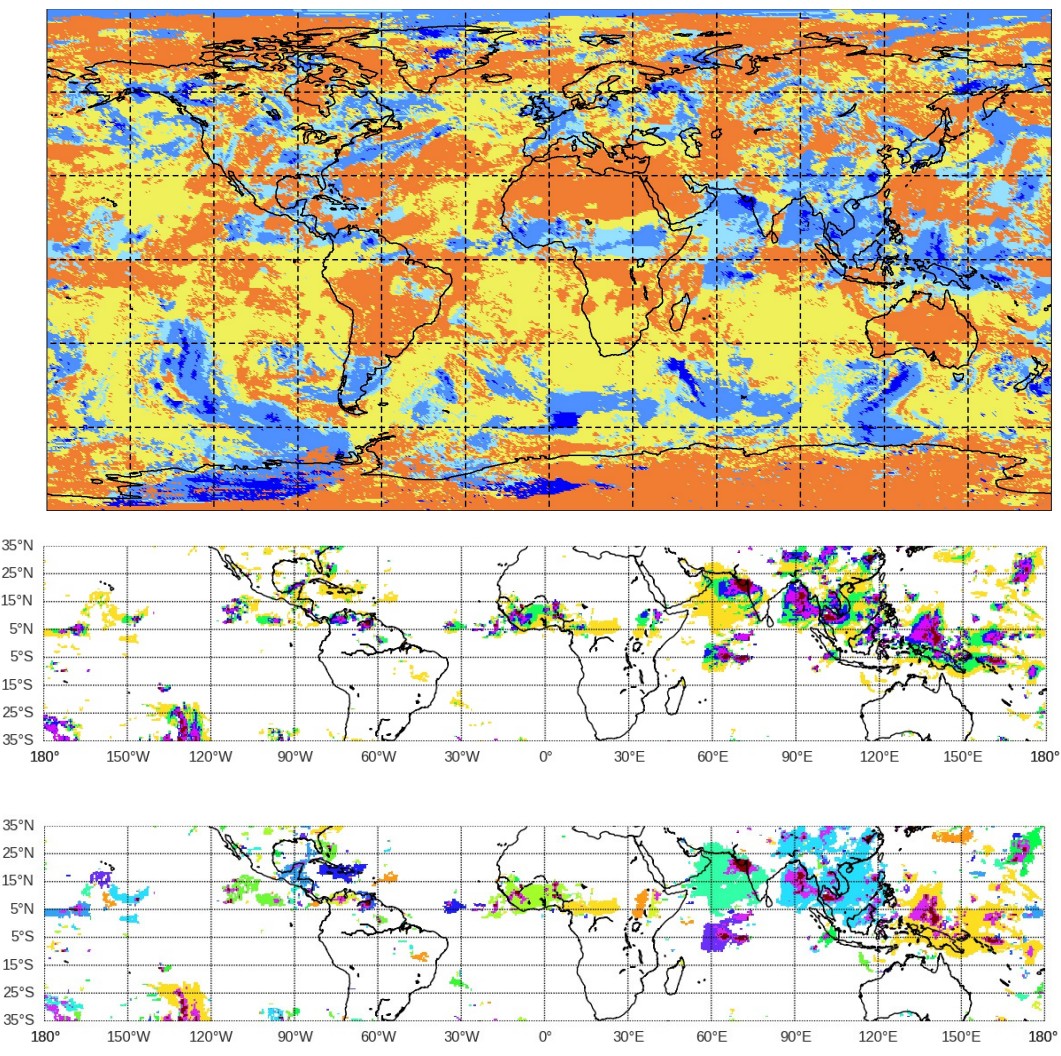

**Figure 4.** Geographic map of AIRS cloud data for 1 July 2007, 1:30 h local time. Top: cloud types, with blue → upper tropospheric clouds (more opaque deeper blue), yellow → mid-level and low clouds and orange → clear sky. Middle: UT clouds for five emissivity classes 0.1, 0.6, 0.8, 0.92, 0.98, 1, represented respectively by yellow, green, blue, magenta, red. Bottom: UT cloud systems, different colours indicate different systems, opaque and convective areas marked with magenta and deep red, respectively.

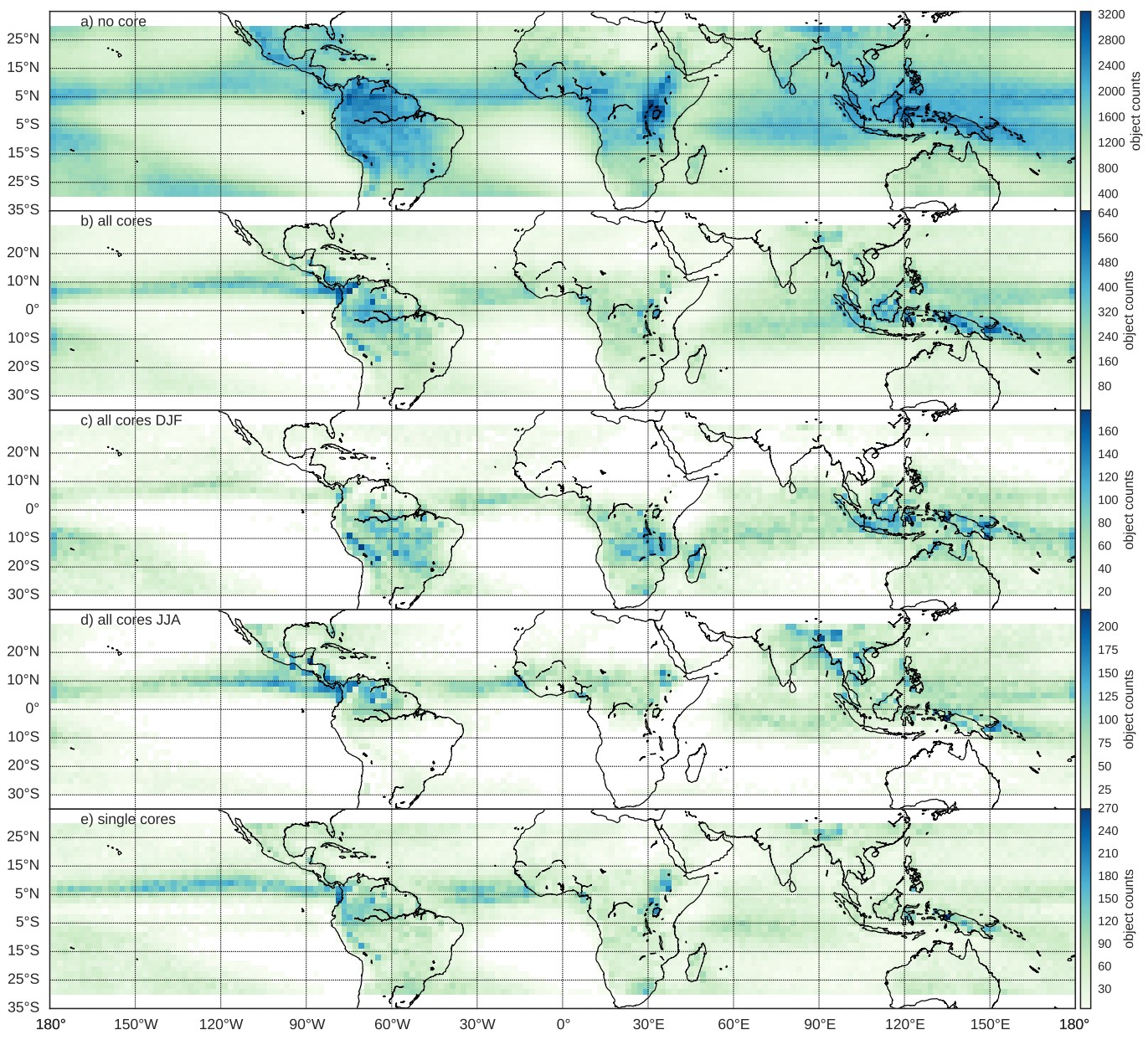

**Figure 5.** Geographic maps of (a) isolated cirrus systems, (b) all convective cores, also separately for (c) boreal winter and (d) boreal summer, and (e) single core convective systems, for the 2003-2015 period of the LMD AIRS cloud climatology. AIRS data, 2003-2015.

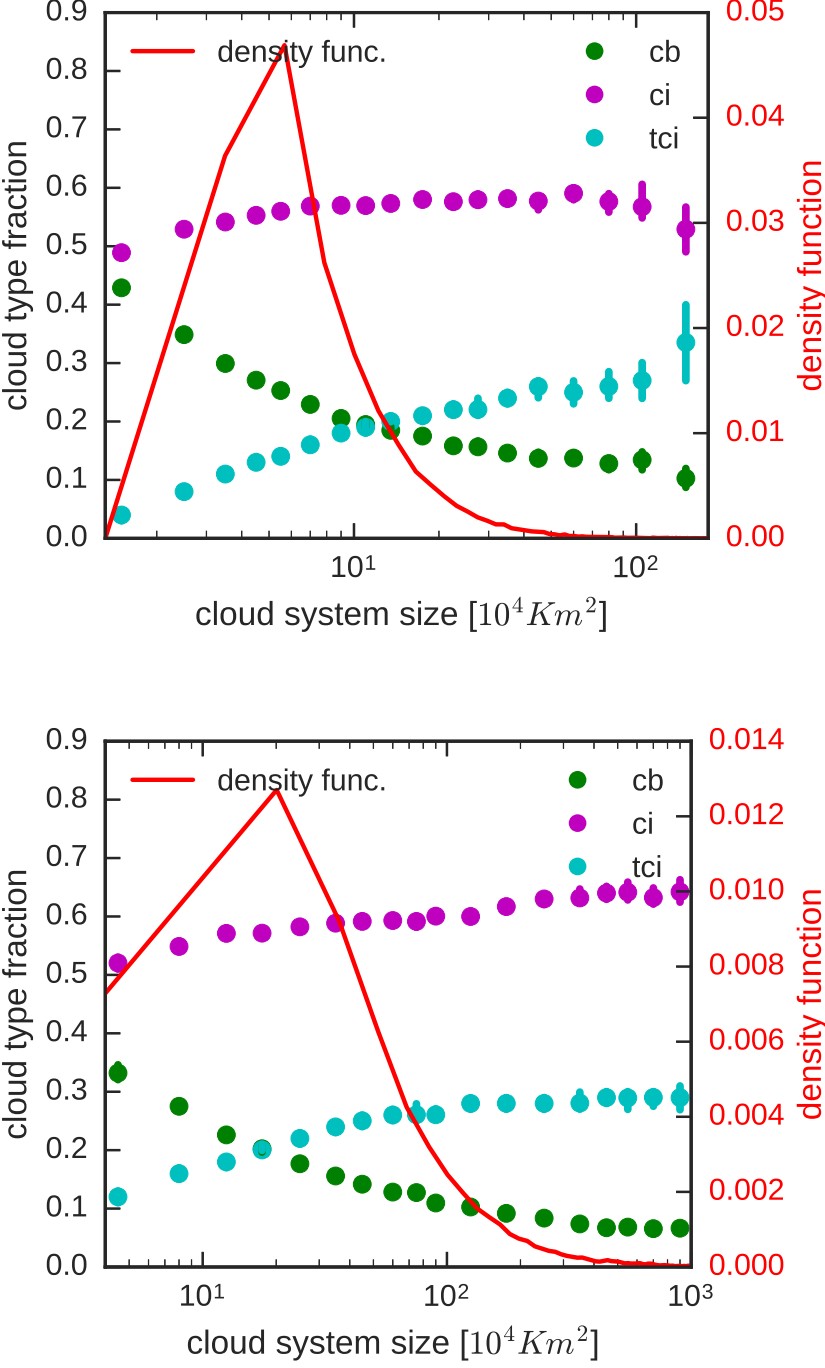

**Figure 6.** Median values and standard errors of fraction of convective core (green), thick (magenta) and thin (cyan) anvil as a function of cloud system size. In red, cloud system size density function distribution. Top: single-core, bottom: multi-core systems. AIRS data, 2003-2015.

**Figure 7.** Convective core fraction kernel density estimate (solid line) and histogram for single (red) and multi (blue) core systems. AIRS data, 2003-2015.

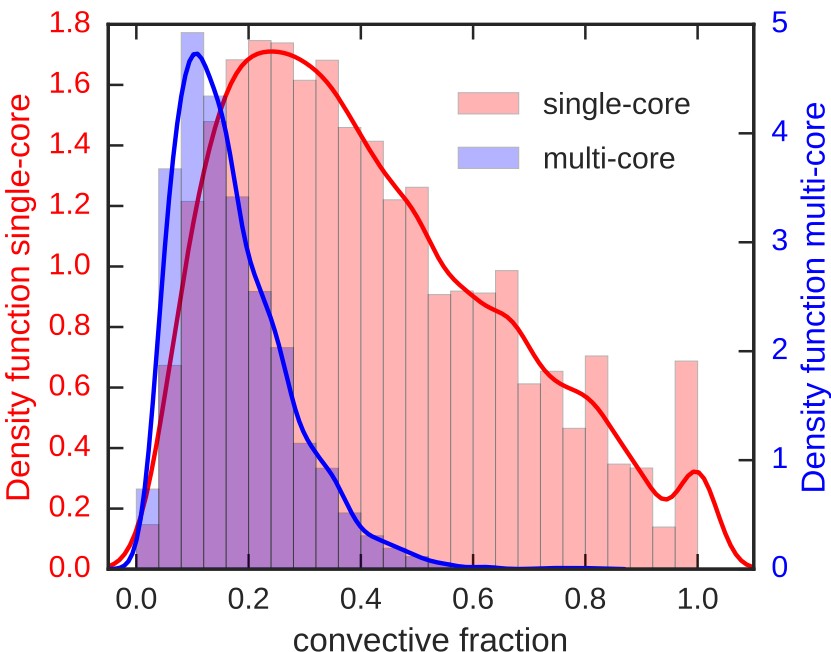

**Figure 8.** Number of single-core cloud systems in each maturity step, separately over ocean and over land and during night (AM) and early afternoon (PM). AIRS data, 2003-2015.

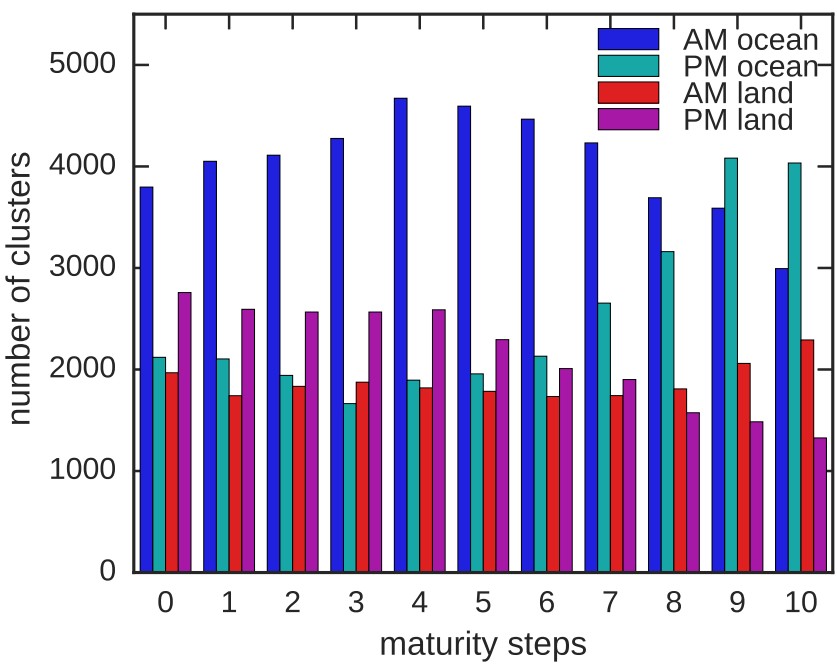

**Figure 9.** Median values and standard errors of physical properties of single-core convective systems for the eleven maturity steps defined by fraction of convective area [1, 0.78, 0.65, 0.55, 0.47, 0.40, 0.34, 0.29, 0.24, 0.19, 0.13, 0.01] , separately over ocean and over land and during night (AM) and early afternoon (PM): a) cloud system size, b) convective core size, c) thin cirrus over cirrus area, d) cloud system average emissivity, e) minimum temperature within convective core, f) average convective core rain rate. a) to e) AIRS data, 2003-2015, f) AIRS and AMSR-E data, 2003-2009.

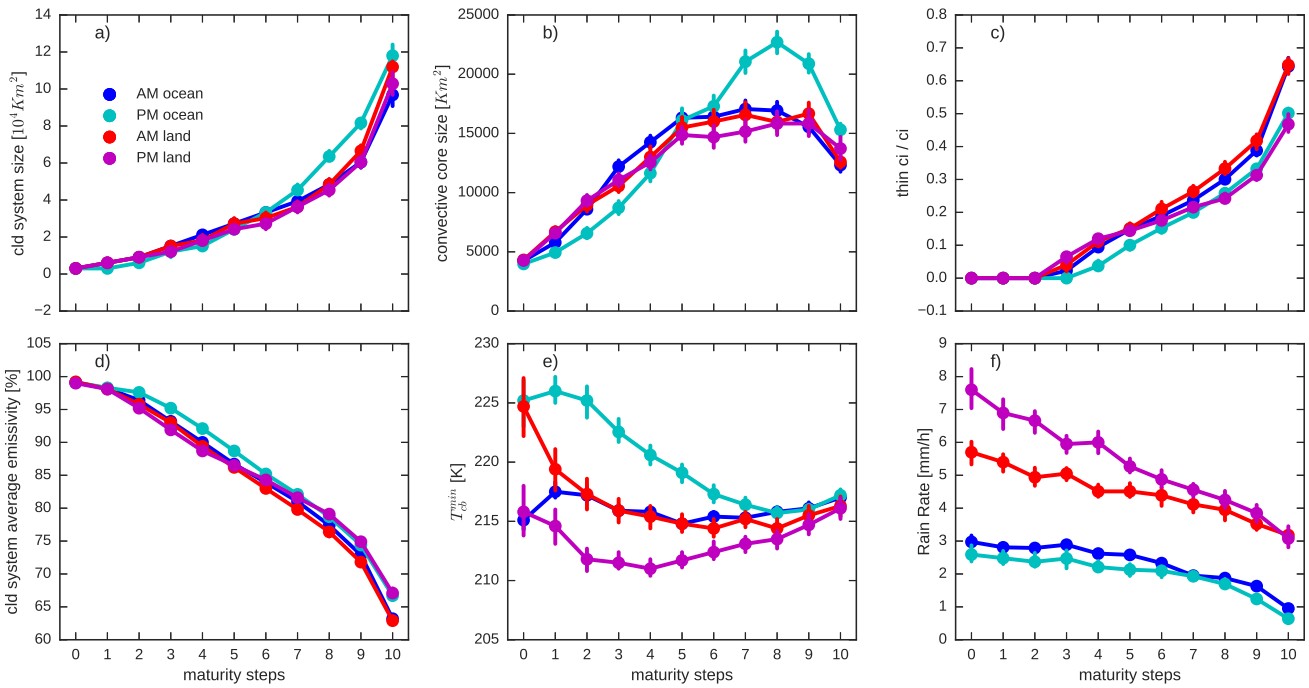

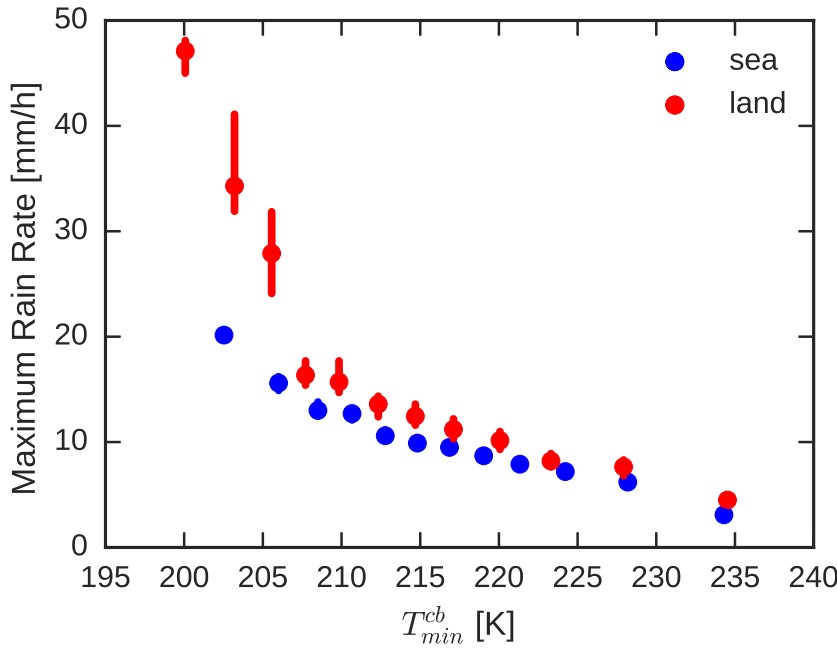

**Figure 10.** Median and standard error of maximum convective core rain rate as a function of minimum temperature within the convective core for mature single-core systems, separately over land (red) and ocean (blue). AIRS and AMSR-E data, 2003-2009.

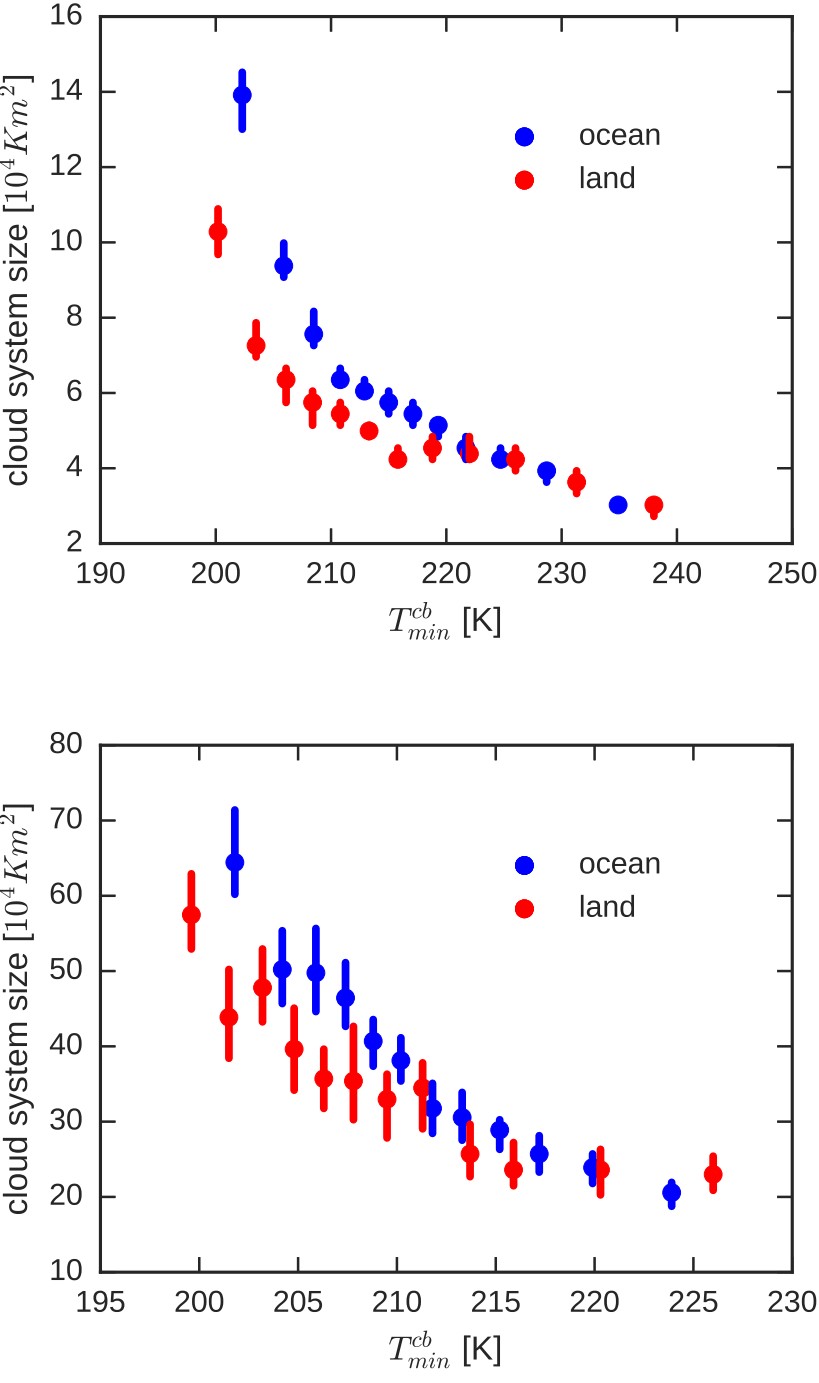

**Figure 11.** Median and standard error of horizontal extent versus minimum temperature within convective core for mature single core (top) and multi-core (bottom) systems, separately over land (red) and ocean (blue). AIRS data, 2003-2015.

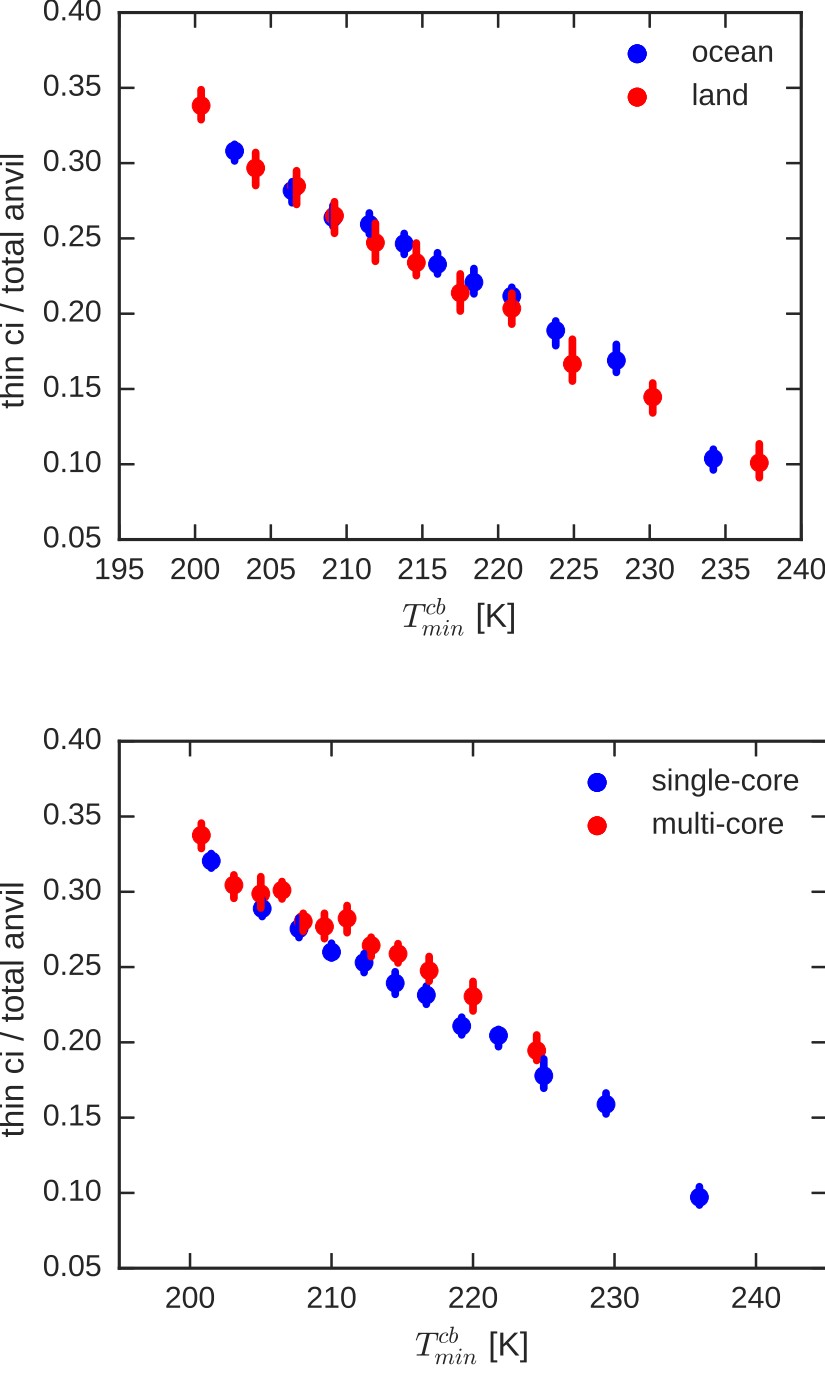

**Figure 12.** Median and standard error of thin cirrus over total anvil area for mature systems as a function of minimum temperature within convective core(s), for single-core systems, separately over land and ocean (top), and separately for single-core (blue) and multi-core (red) systems (bottom).

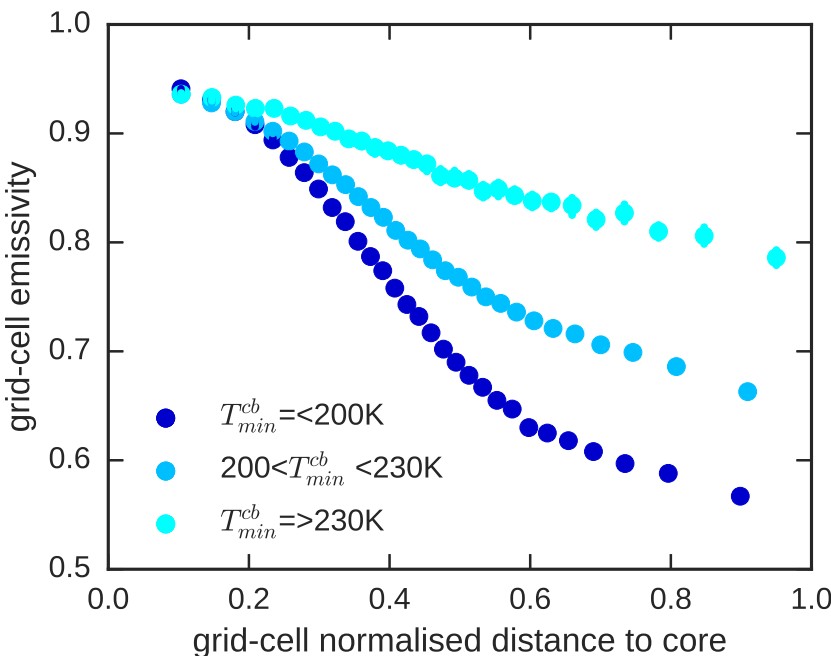

**Figure 13.** Median and standard error emissivity within cloud system as a function of the normalized distance to the convective core. Mature single-core systems are considered for three classes of convective depth represented by intervals in $T_{min}^{cb}$.

|  | isolated cirrus | single core | multi-core |
|---|---|---|---|
| Numb.of systems | >95% | 3% | 1% |
| Coverage | 25% | 10% | 65% |
| Median size | $10^4 \mathrm{Km}^2$ | $10*10^4 \mathrm{Km}^2$ | $200*10^4 \mathrm{Km}^2$ |

**Table 1.** Fraction of occurrence, coverage and median size for isolated cirrus systems, systems with one convective core and with multiple convective cores, over the latitude band 30° N-30° S, annual average over the period 2003-2015.