# Peer review of "Upper Tropospheric Cloud Systems Derived from IR Sounders: Properties of Cirrus Anvils in the Tropics"

_Atmospheric Chemistry and Physics, 2016_

## Referee Comment (RC1) · Anonymous Referee #1 · 3 Nov 2016

The authors use AIRS data and analyzes the horizontal extent of convective and cirrus clouds. The authors grid Level 2 AIRS data in 0.5 by 0.5 degree grids. Cloud type is determined based on the cloud top pressure and emissivity derived from 8 AIRS channels from 11 to 14 microns. Three cloud types, isolated cirrus, and single- and multi-core convective clouds are analyzed in this study. Isolated cirrus and convective systems cover 5% and 15% of the tropical band between 30N to 30S. For convective systems, the areal fraction of the convective core decreases and thin cirrus increases as the system size increase. Earlier studies show that the size of convective systems depend on their life cycle stage. While assuming the areal fraction of convective core relative to the total area of the system, the authors separate single-core convective

systems into eleven intervals of the fraction. The size of core matures and decreases when the stage moves toward dissipation, but thin cirrus area continues increasing throughout the life time. The authors also analyze precipitation derived from precipitation data from microwave sounder AMSR-E. The rain rate averaged over the core area decreases as the systems become more mature. The paper is well written and easy to understand. I only have minor comments and questions to clarify the consistency of their results shown in figures.

Minor comments Page 6 line 29 to 30 Instead of saying "we explore the core fraction follows the evolution of convection life cycle", the authors might want to say that the life cycle state is defined using the core areal fraction. Once the life cycle stage is defined by the fraction, the authors do not need to prove that the fraction follows the evolution of convection, which they really haven't done in Section 3.2, although Figure 7 indicates that it might be the case.

Figures 7a and 7e Generally, the core temperature over land is much colder than that over ocean. But the system size over ocean and land is similar for land and ocean. In addition, the order of the core size at the mature stage step less than 6 is not inverse order of the core temperature. Do you have any explanations of this?

Figures 7b, 7e and 7f The rain rate averaged over the core area almost monotonically decreases with mature stage but the convective core size and minimum temperature within convective core do not. I would expect that the rain rate peaks around a middle stage (perhaps 3 to 5?). Do you have any explanations why the rain rate does not follow the size and minimum temperature of core and it monotonically decreases with maturity steps? Also, is this consistent with Figure 9 showing that the average core rain rate increases with decreasing minimum core temperature?

Page 8 line 18 to 20 This is probably because for a same minimum cloud top temperature, the convective system over ocean is more mature than that over land, hence with a large size and less rain rate, according to figure 7. Is this correct?

---

## Referee Comment (RC2) · Anonymous Referee #2 · 9 Nov 2016

Overview

This manuscript connects properties of tropical cirrus anvils to properties of "convective cores" producing them using AIRS observations. Convective cores are defined as having emissivity values greater than 0.98 based on correlations with AMSR-E rain rate retrievals. Systems with convective cores cover 15% of the area between 30°S and 30°N, while isolated cirrus without cores cover another 5%. Multi-core systems account for 1% of all cirrus systems, but account for 65% of cirrus coverage. Single core system life cycle is estimated using convective area fraction as a proxy for system age. Although land systems produce colder cloud tops and higher rain rates than ocean systems, system size and average emissivity increase similarly as the system

ages. Thin cirrus coverage increases as a fraction of total cirrus coverage (thick + thin) as the system ages. Some differences are apparent for early afternoon and early morning satellite overpasses, presumably because of differences in the probability of life cycle stages at these discrete times. Convective intensity is defined using the minimum retrieved cloud top temperature. As it increases (cloud tops become colder) in mature systems (convective core area fraction between 0.1 and 0.3), system size increases and the thin cirrus area fraction increases as well.

This manuscript presents interesting findings that are worthy of publication, but many of the findings rely on key assumptions that bypass deficiencies of the observations used without exploring their impact on the results and conclusions. The potential impact of these assumptions and deficiencies need to be better assessed, as described further in the major comments below. Satisfactorily addressing the comments will require major revisions, but most of them should be straightforward and hopefully clarify interpretation of the results.

Major Comments

1. Some of the dataset and methodology deficiencies and caveats of the results need to be explained in more detail. For example:

a. A major deficiency of AIRS and AMSR-E is that they only make observations at 2 times of day (∼0130 and 1330 LT). This particularly biases results over land as the average tropical diurnal cycle in deep convection has a strong peak in the late afternoon (∼1600 LT), which is well known from TRMM observations. Furthermore, this diurnal cycle varies by geographical location, so more intense or larger systems are likely favored more in some regions at 0130 or 1330 LT and in others at different times that are not captured by AIRS and AMSR-E.

b. That convective core area fraction is correlated with system life cycle stages is a major assumption. Although single core systems are isolated for analysis, there is no reason to think that some of these systems did not evolve from or into multi-core

systems. If this is the case for a significant fraction of convective systems, then the life cycle stages shown are not representative of typical system evolution, which should not be constrained to single core or multi-core categories for the entire life cycle. It may be the case that this does not heavily impact the statistics shown in Figure 7, but this should be proven to be the case, for example by examining actual life cycles using geostationary satellite data.

2. It is difficult to formulate physical interpretations of "thin" and "thick" anvil cirrus. The distinction seems fairly arbitrary (emissivity greater than or less than 0.5). Since CloudSat and Calipso are also flown in the A-Train satellite constellation, why not show a CloudSat/Calipso cross-section that shows how a typical convective system would be split up into convective core, thick cirrus, and thin cirrus so that the readers can better understand the physical differences between these cloud categories?

3. The definition of "convective core" is different than in most studies of deep convective systems. Typically, this refers to the region with buoyancy driven vertical motions over a relatively deep layer that produces net latent heating throughout most of the tropo-sphere or the region with denser hydrometeors and higher condensates produced via convective motions. The definition used in this manuscript is really just a deep raining region that could be either convective or stratiform. The average AMSR-E rain rates for emissivities between 0.98 and 1 that define convective cores are $\sim$1-3.5 mm/h, which are more consistent with stratiform rain rates than convective rain rates. At the very least, a significant fraction of convective core areas, as they are defined in this study, likely contain stratiform rather than convective precipitation and vertical motions. This should be clarified in the revised manuscript perhaps by renaming the convective cores as deep precipitation cores.

4. In addition to convective intensity, the level of neutral buoyancy for ascending, buoyant air strongly impacts the minimum cloud top temperature. The level of neu-tral buoyancy is likely to be related to the properties of the tropical tropopause transi-tion layer and the cold point tropopause. The tropical tropopause temperature varies

significantly by latitude and season (e.g., Seidel et al. 2010, JGR-Atmospheres, http://onlinelibrary.wiley.com/doi/10.1029/2000JD900837/full; Fueglistaler et al. 2009, Rev. Geophys., http://onlinelibrary.wiley.com/doi/10.1029/2008RG000267/full), which means that using minimum cloud top temperature as a proxy for convective intensity across the entire tropics may introduce time and location biases. In other words, different convective intensities as they are defined in this manuscript may be correlated with specific geographical locations and seasons. This should be fairly straightforward to explore by comparing different latitude bands and seasons with one another.

5. Many of the large cirrus systems will cover a piece of land and ocean. How are these systems assigned to land or ocean categories?

6. The reason that AIRS and AMSR-E are used over other satellite datasets needs to be better explained. How does AIRS improve on what can be retrieved by geostationary satellites regarding convective system cloud properties? It is suggested that it can better distinguish optically thin cirrus clouds from warmer, mid level clouds by decoupling cloud altitude and emissivity, but how is this done? Are the results of this manuscript any different than what has already been learned from geostationary datasets such as ISCCP and active sensors such as CloudSat and Calipso? Briefly putting the results of this manuscript in the context of previous studies in the conclusions would be useful.

7. Just because the cloud top temperature is correlated with the rain rate does not mean that it is a good proxy for convective strength or intensity. In fact, a recent study has shown that the highest rain rates in the tropics may not be associated with the most intense convection based on radar reflectivity echo tops (Hamada et al. 2015, Nature Communications, http://www.nature.com/articles/ncomms7213?WT.ec_id=NCOMMS-20150225). Of course, this is using the traditional definition of convective intensity, which typically refers to the updraft vertical velocity magnitude. It has been known for some time though that convection need not be intense to reach the tropopause (e.g., Zipser 2003, Meteorological Monographs, http://link.springer.com/chapter/10.1007/978-1-878220-63-9_5). Therefore, I recom-

mend removal of the "convective intensity" terminology because it is being very loosely, which leads to confusion. Why not simply refer to the "convective depth" instead? It could still be pointed out that it is positively correlated with rain rate.

8. On page 8, line 1-2, it is stated that minimum brightness temperature has been shown to be a more skillful proxy to describe convective intensity compared to the radar echo height based on Jiang (2012), but this is not what is concluded in Jiang (2012). Jiang (2012) states that minimum infrared brightness temperature in the inner core of tropical cyclones is a better indicator of tropical cyclone rapid intensification than other proxies for convective intensity. It is also problematic that this study, which focuses on tropical cyclones, is being used in the manuscript as representative of all tropical MCSs. In fact, it is well known that for a given minimum infrared brightness temperature, convective intensity is far stronger over land than ocean, which is reflected in far different reflectivity profiles, microwave brightness temperatures, and lightning flash rates, which are the traditional measures of convective intensity (e.g., Zipser et al. 2006, BAMS, http://journals.ametsoc.org/doi/pdf/10.1175/BAMS-87-8-1057; Liu et al. 2007, J. Climate, http://journals.ametsoc.org/doi/full/10.1175/JCLI4023.1; many others). Minimum infrared brightness temperature is assuredly not the most skillful proxy for convective intensity when used across the entire tropics.

9. Is Figure 1 just one example of many possibilities or is it an average relationship? Does a pair of given brightness temperature and emissivity values always produce the same retrieved cloud top temperature? For example, will a brightness temperature of 260 K and emissivity of 0.6 always produce a cloud top temperature of 230 K? More information on how cloud top temperature is retrieved and limitations of the retrieval would be helpful.

10. Possible inconsistencies between Figure 7 and Figures 10 and 12 need to be explained. For example, in Figure 7a, cloud system size increases from stage 6 to stage 9, but in Figure 7e, the minimum cloud top temperature increases from stage 6 to stage 9 (during system maturity when convective area fraction is between 0.1

and 0.3), so as minimum cloud top temperature increases, system size increases. However, Figure 10 shows system size decreasing with increasing minimum cloud top temperature, which is the opposite relationship. Similarly, Figure 12 shows that the thin cirrus anvil area fraction decreases with increasing minimum cloud top temperature, but Figure 7c and 7e show that thin cirrus anvil area fraction increases with increasing minimum cloud top temperature, which is the opposite relationship.

11. Are the results in Figure 14 a result of different life cycle stages of the single core systems or do these differences also exist for a given life cycle stage (indicating differences in the life cycles of systems of varying convective depth)? Clarification here would provide valuable insight into the results.

Minor Comments

1. I suggest changing "build" on page 1, line 23 to "are part of" since the clouds are primarily a function of the convection rather than the other way around.

2. On page 2, line 5, there appears to be a missing word after "MCS's".

3. The data is gridded at a resolution of 0.5°, but it seems that the distribution of cloud types defined at the native measurement resolution in each grid box is use for all of the figures. Is this correct?

4. The gap between orbits is largest at the equator. Please state the width of the gap and scan at the equator on page 4, line 19.

5. Which latitude band is used for most of the figures? This should be mentioned in Section 2.

6. Please capitalize "south" on page 5, line 26.

7. Remove "s" from "includes" on page 6, line 9.

8. Insert "do" after "dissipation" on page 6, line 24.

9. Change "is" to "are" on page 7, line 18.

10. Change "on" to "in" on page 7, line 22.

11. Remove "or" on page 7, lines 24 and 25.

12. Change "signal" to "reflectivity" and "or brightness temperature" to "and microwave or infrared temperatures" on page 7, line 25.

13. Insert "cloud top" before "temperature" on page 7, line 31.

14. Does the resolution of the minimum retrieved cloud top temperature used in analyses change based on distance from nadir?

15. Please clarify what is meant by "rain detection offset over land" on page 8, line 8.

16. Change "is" to "it" on page 8, line 31.

17. A citation is needed for the statement that convective intensity will increase in a warming climate on page 8, lines 32-34.

18. Insert "of" after "years" on page 9, line 2.

19. Why do the distributions in Figure 6 go less than 0 and greater than 1 when convective fraction cannot be less than 0 or greater than 1?

20. Are the bars in Figures 7 and 9-13 standard errors of the mean? If so, please state that and include the sample sizes used to make the figures.

---

## Author Comment (AC1) · 12 Jan 2017

**Overview**

The authors use AIRS data and analyzes the horizontal extent of convective and cirrus clouds. The authors grid Level 2 AIRS data in 0.5 by 0.5 degree grids. Cloud type is determined based on the cloud top pressure and emissivity derived from 8 AIRS channels from 11 to 14 microns. Three cloud types, isolated cirrus, and single- and multi-core convective clouds are analyzed in this study. Isolated cirrus and convective systems cover 5% and 15% of the tropical band between 30N to 30S. For convective systems, the areal fraction of the convective core decreases and thin cirrus increases as the system size increase. Earlier studies show that the size of convective systems depend on their life cycle stage. While assuming the areal fraction of convective core relative to the total area of the system, the authors separate single-core convective systems into eleven intervals of the fraction. The size of core matures and decreases when the stage moves toward dissipation, but thin cirrus area continues increasing throughout the life time. The authors also analyze precipitation derived from precipitation data from microwave sounder AMSR-E. The rain rate averaged over the core area decreases as the systems become more mature. The paper is well written and easy to understand. I only have minor comments and questions to clarify the consistency of their results shown in figures

We kindly thank the reviewer for his comments. Before addressing them we would like to summarize the major changes made in the new version of the article as has been required during the revision process. These changes mainly aimed to: a) make the motivation of this work clearer, b) give further explanations on the AIRS InfraRed sounder data retrieval and their advantages, c) clarify the usage of convective fraction as a maturity index, d) introduce a discussion on the various existing convective proxies e) provide more interpretations to the results.

More in detail, it was made clearer that the 'life-cycle' section targeted to the definition of maturity stage necessary for exploring properties of mature convective systems. For mature systems it is more appropriate to refer to 'convective depth' (given by cloud top height or temperature) rather than 'convective intensity', as the latter is related to dynamical conditions which has not been profoundly investigated in this article. In order to make clearer our purpose we created a new section (4) out of section 3.3, where the discussion on convective intensity and depth is conducted.

Main changes per section:
**1.0** Introduction: Clarify motivation and give context for this article. Added a paragraph introducing the convective intensity/ strength and depth discussion.

**2.1:** ADDED two paragraphs giving more details on the retrieval methodology

**2.2:** ADDED: A) quartile bands in figures 1 and 3 (see new document). B) in plot 3 vertical winds at 500hPa from ERA Interim, to show that even with a broad distribution one observes larger winds for more opaque clouds. C) a figure (Fig.2) showing the relation between the cloud temperature ($T_{cld}$), the infrared brightness temperature ($T_B$) and emissivities. D) a middle panel in figure 4 in which UT cloud system emissivity is shown categorized in 5 classes.

**3.1:** ADDED in figure 5, 3 panels (c,d,e) showing  c)  all DJF cores, d) JJA cores and e) all cores of single-core systems

**3.2:** ADDED in figure 7 the histogram.
 We also inversed the order of figures 8 and 9 to better fit the text in which now there is a discussion on the diurnal variation and the life-time duration of convective systems, to support our point that even with only two measures per day,

we will capture systems in different maturity phases. More details on the comparison of our results with previous studies (Fiolleau and Roca, 2013 and Machado, 1998) are given.

**4.0:** Former section 3.3 is now section 4. Added two paragraphs introducing the convective intensity/strength and depth, how these can be measured, and how this can be done. With AIRS alone it is more appropriate to refer to convective depth as this is linked to the altitude of the system and therefore to its Temperature, a variable available in our data.

**5.0:** Re-worked on the conclusion so that it reflects all the modifications discussed above.

**Comments**

Minor comments Page 6 line 29 to 30 Instead of saying "we explore the core fraction follows the evolution of convection life cycle", the authors might want to say that the life cycle state is defined using the core areal fraction. Once the life cycle stage is defined by the fraction, the authors do not need to prove that the fraction follows the evolution of convection, which they really haven't done in Section 3.2, although Figure 7 indicates that it might be the case.

In the phrase '*we explore whether they follow an evolution pattern which corresponds to different life cycle stages* ', the 'they' refers to the physical properties. Indeed the text wasn't very clear, so we have added more text explaining why we use the convective fraction as a maturity indicator and figure 9 has been better explained.

 Figures 7a and 7e: Generally, the core temperature over land is much colder than that over ocean. But the system size over ocean and land is similar for land and ocean. In addition, the order of the core size at the mature stage step less than 6 is not inverse order of the core temperature. Do you have any explanations of this?
 Figures 7b, 7e and 7f: The rain rate averaged over the core area almost monotonically decreases with mature stage but the convective core size and minimum temperature within convective core do not. I would expect that the rain rate peaks around a middle stage (perhaps 3 to 5?). Do you have any explanations why the rain rate does not follow the size and minimum temperature of core and it monotonically decreases with maturity steps? Also, is this consistent with Figure 9 showing that the average core rain rate increases with decreasing minimum core temperature?

We use this figures essentially to determine the maturity of the cloud system: While the horizontal extent of the convective core increases until it reaches a plateau, which corresponds to a convective fraction between about 0.1 – 0.3, and therefore indicates the convective fraction as a proxy of maturity, with emissivity, total cloud system size and rain rate behaving as one would expect (for instance the rain rate decrease was also seen by Fiolleau and Roca (2013), the min temperature of the convective core is the only variable which has no clear behaviour. It should be stressed that these are median values per maturity step and each of these values corresponds to a distribution. When considering specific regions, like the three land regions and three ocean regions discussed in (Liu and Zipser, 2008), the behaviour is similar as in Figure 7, with similar Tcbmin over all maturity steps in the less convective ocean regions and with slightly colder Tcbmin values when the convective fraction is larger for the other regions (Fig 1 in supplement). However, all minimum temperatures of the convective cores seem to converge towards a plateau for the mature and dissipating convective systems. We are interested to study relationships between anvil properties and convection when the systems are mature.
We have added this discussion into the manuscript and we have changed average rain

rate vs TCbmin to maximum rain rate vs TCbmin, which describes better the convective part of the rain, while the other one includes stratiform rain.

Page 8 line 18 to 20: This is probably because for a same minimum cloud top temperature, the convective system over ocean is more mature than that over land, hence with a large size and less rain rate, according to figure 7. Is this correct?

Plots 11 and 12 (numbering in the new document) are for mature convective systems (0.1<cb_frac<0.3) for both land and ocean. To make sure that the differences observed between land and ocean are not due to a too broad definition of maturity in Fig.1 is presented the correlation between the size and the minimum temperature if one restricts the maturity definition between 0.2 and 0.25. The distribution is almost identical with the one shown in the paper, indicating that this difference is probably linked to different convection properties between land and ocean rather than to a different maturity stage.
Indeed, oceanic convective systems of a similar convective depth as continental systems have a larger size with less intense convective rain, a behavior significantly enhanced for systems with an important convective depth. This difference in structure was already pointed out in earlier studies (e.g. Liu et al., 2007). Furthermore, Liu et al. (2007) have shown that tropical continental mesoscale convective cloud systems are in general smaller in size than oceanic systems, though the vertical updraft and horizontal extent of the convective cores are in general larger, while their convective depth is similar. Their ice water path is also larger than the one of oceanic systems, which is caused by different microphysics between land and ocean (Sohn et al., 2015).

We have added a long discussion in Section 4.

[Figure]

**Figure 1.** Total system size versus convective core minimum temperature for mature single core convective systems having a convective fraction between 0.2 and 0.25

[revised manuscript text omitted]

---

## Author Comment (AC2) · 12 Jan 2017

**Overview**

This manuscript connects properties of tropical cirrus anvils to properties of "convective cores" producing them using AIRS observations. Convective cores are defined as having emissivity values greater than 0.98 based on correlations with AMSR-E rain rate retrievals. Systems with convective cores cover 15% of the area between 30S and 30N, while isolated cirrus without cores cover another 5%. Multi-core systems account for 1% of all cirrus systems, but account for 65% of cirrus coverage. Single core system life cycle is estimated using convective area fraction as a proxy for system age. Although land systems produce colder cloud tops and higher rain rates than ocean systems, system size and average emissivity increase similarly as the system ages. Thin cirrus coverage increases as a fraction of total cirrus coverage (thick + thin) as the system ages. Some differences are apparent for early afternoon and early morning satellite overpasses, presumably because of differences in the probability of life cycle stages at these discrete times. Convective intensity is defined using the minimum retrieved cloud top temperature. As it increases (cloud tops become colder) in mature systems (convective core area fraction between 0.1 and 0.3), system size increases and the thin cirrus area fraction increases as well.

This manuscript presents interesting findings that are worthy of publication, but many of the findings rely on key assumptions that bypass deficiencies of the observations used without exploring their impact on the results and conclusions. The potential impact of these assumptions and deficiencies need to be better assessed, as described further in the major comments below. Satisfactorily addressing the comments will require major revisions, but most of them should be straightforward and hopefully clarify interpretation of the results.

First of all we thank the reviewer for his thoughtful comments which helped immensely to improve the manuscript. Before addressing one by one the comments, we would like for the sake of clarification to summarize why we intended to build this data base of UT cloud systems using AIRS cloud properties. One should keep in mind these points when reading the answers to the questions. This short introduction also provides partial answers to questions 2, 6 and 9.

Our AIRS cloud retrieval makes use of eight spectral channels sounding along the 15 micron $CO_2$ absorption band, providing cloud pressure $p_{cld}$ and emissivity $\varepsilon_{cld}$ of a single cloud layer which corresponds to the uppermost cloud layer in the case of multi-layer clouds (Stubenrauch et al. 2010). The method takes into account the vertical weighting of the different channels, the growing uncertainty in the computation of $\varepsilon_{cld}$ with increasing p and uncertainties in atmospheric profiles. The main advantage of IR sounders is their reliable determination of $p_{cld}$ and $\varepsilon_{cld}$ for cirrus clouds down to an IR emissivity of 0.1 (corresponding to a visible optical depth of 0.2), day and night. Once $p_{cld}$, and $\varepsilon_{cld}$ are retrieved by a $\chi^2$ method (Stubenrauch et al. 1999), cloud temperature $T_{cld}$ is determined from $p_{cld}$, by using the AIRS temperature profile. The construction of the UT cloud systems is undertaken in two steps: first adjacent measurements with similar $p_{cld}$ (up to 250 hPa underneath the tropopause) are used to compose these systems, and then we use $\varepsilon_{cld}$ to distinguish between convective cores, thick anvil and thin anvil.

Compared to methods which use cold IR brightness temperature $T_B$ to track convective cores, as is done using geostationary satellite imagery, our method has the key advantage that we can reliably distinguish between semi-transparent cirrus and opaque high clouds and deduce for both cloud types a reliable height. This allows for the first time to account for semi-transparent cirrus in the convective cloud systems, the latter are very important to understand radiative effects.

The motivation of this article is to present this data base, which, coupled with other data, will provide observational metric for a better understanding of the interconnection between tropical convection and the heating induced by the

outflowing anvils.

Indeed, there are also drawbacks to this data set, like the fact that AIRS data are only available twice daily and with AIRS alone we can explore only a proxy for convective depth which is not identical to convective intensity, the latter identified by the dynamics, and therefore we will complete this data base in the future with complementary available data. However, already with the AIRS data alone we are able to explore the anvil properties in relation to the convective depth.

We have added these explanations in the introduction and methodology sections of our manuscript and hope that this is now much more understandable for readers who are not so familiar with IR sounder data.

**Major Comments**

**1.** Some of the dataset and methodology deficiencies and caveats of the results need to be explained in more detail. For example:

**a.** A major deficiency of AIRS and AMSR-E is that they only make observations at 2 times of day (? 0130 and 1330 LT). This particularly biases results over land as the average tropical diurnal cycle in deep convection has a strong peak in the late afternoon (1600 LT), which is well known from TRMM observations. Furthermore, this diurnal cycle varies by geographical location, so more intense or larger systems are likely favored more in some regions at 0130 or 1330 LT and in others at different times that are not captured by AIRS and AMSR-E.
b. That convective core area fraction is correlated with system life cycle stages is a major assumption.

Indeed, we have only two measurements per day and temporal (and geographical) variabilities exist. It is known that the large-scale atmospheric dynamics and radiative processes strongly affect the life cycle of deep convective systems in the tropics. While over land a maximum of precipitation is expected in the late afternoon, over ocean tropical convection occurs a few hours before sunrise (with a very broad peak) as has been shown for example by Liu and Zipser (2008) or by Yamamoto et al. (2007). This means that the AIRS data are collected a few hours earlier than the respective day and night maxima in tropical convection. Therefore the analyzed convective cloud systems might be somewhat weaker than what would have been observed a few hours later in the local day.

 However, in particular organized convection has often a life time longer than 24 hours which makes it possible to explore statistically these convective systems; it has already been demonstrated in previous studies using satellite data with better temporal resolution (geostationary imagery data, Machado et al 1998 and 2003, Futyan and Del Genio 2007) or with varying observation time (TRMM,Fiolleau and Roca 2013 etc) that the largest systems have the longest life cycle, up to 30h. Therefore even with only two measurements per day we should be able to observe systems in different phases of their life cycle.

Our article is not focused on studying the life cycle, but we use as proxy of maturity stage the **fraction of convective core horizontal extent with respect to the horizontal extent of the whole cloud system** to find a way to isolate relatively mature convective systems, so that we can explore their relationship between convective depth and anvil properties. T**his variable has been proven to be an indicator of convective cloud maturity (and hence for following the life cycle when the temporal resolution is good enough) in studies using IR imagery of geostationary satellites (Machado et al 1998), TRMM (Fiolleau and Roca 2013), as well as using CloudSat radar (Bacmeister and Stephens 2011**). When using this proxy, one observes in figure 8, as expected, slightly more 'developing' systems over land at PM while more dissipating systems over ocean at the same time, and vice versa.

When exploring further this proxy with our data, we observed that, statistically the evolution of the properties of the convective systems seem to be consistent with what one would be expect: like decreasing $\varepsilon_{cld}$, increasing of convective core size until maturity and then decrease (Fig. 9). In particular, the average rain rate in the convective core decreases as the system gets older, in agreement with (Fig. 5 of Fiolleau et Roca 2013). Therefore we felt confident to select relatively mature convective systems with a convective core fraction of about 0.1 – 0.3. It is true that our definition of convective core, through $\varepsilon_{cld}$ close to 1, might include a stratiform rain fraction (as in studies using $T_B$ as in Machado et al. 1998), but again this selection is used just to choose convective systems with similar maturity in order to study anvil properties as function of convective depth.

The cloud system size increases with decreasing convective function, something expected as the detrained anvil increases as the system gets older, it should be stressed that we do not capture the anvil shrinking as shown in Fig. 9a of (Machado et at 1998), most likely, because Machado et al. studied only the thicker anvils and we see that the thinner anvil part increases towards dissipation. We also consider systems having at least 1 convective grid with emissivity above 0.98 and therefore the system is not captured in its advanced dissipation. This is a weakness of our analysis for the life-cycle exploration which has been **explained in the text**.

To study land/ocean differences we perform the analysis separately. In the present paper we did not foresee a further division of the statistics per region because we are first interested in 'global' behaviors and relationships, and we think that differences in regions can be probably explained by differences in additional dynamical variables which we foresee to add when ERA5 data (with a better horizontal and temporal resolution) are getting available.

[Figure]

**Figure 1.** Key relations in latitudinal bands of $20^{\circ}$ for DJF and JJA for mature single core systems (top panel) and in particular regions (bottom panel). Right: thin cirrus over anvil versus convective core minimum temperature. Left: total convective system size versus convective core minimum temperature.

Still, for addressing the question on the geographical and seasonal variability, and to prove that our findings do not depend on these variabilities, we present here, in Fig.1, the main correlation plots of the paper for mature single-core systems, in the top panel separately for DJF and JJA over three latitudinal bands (-30<lat <-10, -10<lat<10 and 10<lat<30), and in the bottom panel for regions define as in Liu and Zipser 2008. These plots show that the fraction of thin cirrus over the total anvil area as well as the horizontal extent of the convective systems increase with decreasing minimum convective core temperature for all configurations even though the slope might slightly differ. Therefore, the main findings of this article are  robust as well reproduced for all latitudinal-band, seasonal and regional configurations : a) larger systems penetrate deeper into the troposphere (in agreement with other publications, like Rossow and Pearl 2007), b) the ratio of thin cirrus over total anvil is higher for colder mature systems.

 Although single core  systems  are  isolated  for analysis,  there  is no reason to think that some of these systems did not evolve from or into multi-core systems.
 If this is the case for a significant fraction of convective systems, then the life cycle stages shown are not representative of typical system evolution, which should not be constrained to single core or multi-core categories for the entire life cycle. It may be the case that this does not heavily impact the statistics shown in Figure 7, but this should be proven to be the case, for example by examining actual life cycles using geostationary satellite data.

Indeed, this 'transition' between cloud system types might happen. However, a single core which evolves as multi-core due to the emergence of a new 'tower' in its vicinity is not be considered in the determination of maturity; the other way around (multi-core to single core) is more complicated to completely exclude these events. One motivation for this article was to find an observational metric which can be used for the evaluation of the simulation of detrainment processes. Therefore we wanted first to study the cleanest single-core sample possible: all single-core systems having a second opaque area (0.9<eps<0.98) are excluded for making the plots, **this information has been added to the paper text**.

 To asses the potential bias in the life cycle study due to this 'migration', we included in the statistics also multicore systems. It should be stressed, that, as can be seen from figure 7 in our article, the majority of multi-core systems has a convective fraction between 1% and 40% and therefore only maturity steps between step 5 to 10 intervals will be affected. Figure 2 presents the results if single and multi-core systems are put together; the behaviour is very similar.

[Figure]

**Figure 2.** Median cloud system size (left), ratio of thin cirrus versus cirrus anvil (middle) and convective core average rain rate (right) as a function of maturity steps, including single and multi-core systems.

**2.**  It is difficult to formulate physical interpretations of "thin" and "thick" anvil cirrus. The distinction seems fairly arbitrary (emissivity greater than or less than 0.5).  Since CloudSat and Calipso are also flown in the A-Train satellite constellation, why not show a CloudSat/Calipso cross-section that shows how a typical convective system would be split up into convective core, thick cirrus, and thin cirrus so that the readers can better understand the physical differences between these cloud categories?

There are mainly three reasons why we selected 0.5: a) below this threshold no rain is observed at all b) this threshold has already been used to define thin cirrus in earlier Infrared sounder analyses, c) as shown in section 2.2 all studies using IR brightness temperatures exclude clouds with an emissivity below 0.6, meaning about 30% of the horizontal extent. Therefore we wanted to use a threshold indicative of the difference between IR brightness temperature analysis and our analysis.
For illustration, Fig.3 shows a geographical map of a day scene of AIRS UT cloud systems, distinguishing 5 classes in emissivity.

[Figure]

**Figure 3. Geographical map of** AIRS UT cloud emissivity for the 1[st] of July 2007 AM, with each color indicating an emissivity class: **ε>0.98, ε: 0.92-0.98 , ε: 0.80-0.92, ε:0.60-0.80, ε:0.10-0.60**.

A recent collaboration with H. Takahashi at JPL has led to combine our AIRS convective cloud systems with those determined from CloudSat (Takahashi and Luo 2014). The Figure 4  presents recent results which have been presented at the last GEWEX Cloud and Assessment Panel meeting. So far it was shown that there is a good correlation between AIRS and CloudSat cloud height; AIRS adds to the horizontal

dimension and AIRS extends the convective systems beyond an emissivity of 0.5. In a next step we will compare the relationships between the different proxies of convective intensity and depth, which will be published in a separate paper.

[Figure]

**Figure 4. Slide presented in the** GEWEX Cloud and Assessment Panel meeting (November 2016) **showing the preliminary results of the ongoing collaboration with H.Takahashi for collocating our AIRS objects with CloudSat Objects.** First row**:** two examples of the horizontal view of the AIRS convective cloud systems, with colors showing the grid emissivity and the corresponding CloudSat track in purple (upper panel)) and the CloudSat radar reflectivity profile (bottom panel) . Second row left: correlation of AIRS temperature with CloudSat height. Second row right: distribution of cloud emissivities of AIRS cloud systems when collocated with CloudSat (upper panel), distribution of all cloud emissivities of AIRS cloud systems.

**3.** The definition of "convective core" is different than in most studies of deep convective systems. Typically, this refers to the region with buoyancy driven vertical motions over a relatively deep layer that produces net latent heating throughout most of the troposphere or the region with denser hydrometeors and higher condensates produced via convective motions. The definition used in this manuscript is really just a deep raining region that could be either convective or stratiform. The average AMSR-E rain rates for emissivities between 0.98 and 1 that define convective cores are ?1-3.5 mm/h, which are more consistent with stratiform rain rates than convective rain rates. At the very least, a significant fraction of convective core areas, as they are defined in this study, likely contain

stratiform rather than convective precipitation and vertical motions.  This should
be clarified in the revised manuscript perhaps by renaming the convective cores
as deep precipitation cores.

Indeed, in our statistical analysis we assumed that the probability that an opaque
core within an UT cloud system is linked to convection is high. Our comparison with
precipitation showed a positive correlation between cloud emissivity and
precipitation. The latter might still be of stratiform origin, as it can be seen
from the updated figure 3 in the manuscript of the average rain rate (RR) and Max
RR (within 0.5° grid) versus emissivity where the quartile bands have been added:
there is a probability that even with an emissivity of 1, average rain in the 50 km
grid is below 1 mm/h, but  the maximum RR quartile is at 6mm/h. The link with
convection can only be shown through vertical updraft. Vertical winds from ERA
Interim data are only available at a spatial resolution of 0.75 and temporal
resolution of 6h, this is why we didn't include them in the analysis as 2
interpolations (temporal and spatial) are needed and therefore dilute the
information for their estimation. Though, we included now in Figure 3 also the
median vertical wind (at 500 hPa) and its quartiles. The figure shows that UT
opaque clouds are in general also linked to stronger updrafts.
We did not rename 'convective cores' to 'deep precipitation cores', since this is
also not true in all cases. However, we added a discussion of the reliability of
the opacity proxy for convective cores in the article. Moreover, for consistency,
we have replaced in figure 10 average rain rate inside the convective core with the
maximum rain rate, as this quantity is more representative of the convection while
the average value might also consider the stratiform rain. One observes that over
land the slope of the Maximum rain rate with the convective depth is significantly
steeper when the system is colder.

**4.**  In addition to convective intensity, the level of neutral buoyancy
for  ascending, buoyant air strongly impacts the minimum cloud top temperature.
The level of neutral buoyancy is likely to be related to the properties of the
tropical tropopause transition layer and the cold point tropopause.  The tropical
tropopause temperature varies significantly  by  latitude and  season (e.g.,
Seidel  et  al.    2010,  JGR-Atmospheres,
http://onlinelibrary.wiley.com/doi/10.1029/2000JD900837/full; Fueglistaler et al.
2009, Rev.   Geophys.,
http://onlinelibrary.wiley.com/doi/10.1029/2008RG000267/full),  which means that
using minimum cloud top temperature as a proxy for convective intensity across the
entire tropics may introduce time and location biases.  In other words, different
convective intensities as they are defined in this manuscript may be correlated
with specific geographical locations and seasons. This should be fairly
straightforward to explore by comparing different latitude bands and seasons with
one another.

The plots shown for addressing question 1a are also the answer for question 4: we
see that even though there are seasonal and geographical variabilities, i. e.
TCbmin is slightly shifted towards higher values (lower systems) for regions
outside the ITCZ and vice versa, in all configurations the 'pattern' is the same
for mature systems:  larger systems have penetrated deeper in the troposphere.

**5.**  Many of the large cirrus systems will cover a piece of land and ocean.  How
are these systems assigned to land or ocean categories?

In order to categorize a system as 'land' or 'ocean' we use the fraction of 'land'

convective grids (if >=0.5 → land system, if < 0.5 → ocean system). It should be stressed that less than 5% of the total single-core statistics has a land fraction between 0.2 and 0.8 and thus we expect the separation between land and ocean systems to be accurate.  For multi-core systems land fraction is computed the same way, however, as systems are large, this is just an approximation.

**6.** The reason that AIRS and AMSR-E are used over other satellite datasets needs to be better explained. How does AIRS improve on what can be retrieved by geostationary satellites regarding convective system cloud properties? It is suggested that it can better distinguish optically thin cirrus clouds from warmer, mid level clouds by decoupling cloud altitude and emissivity, but how is this done?

Are the results of this manuscript any different than what has already been learned from geostationary datasets such as ISCCP and active sensors such as CloudSat and Calipso? Briefly putting the results of this manuscript in the context of previous studies in the conclusions would be useful.

We have given an explanation at the beginning of our replies and have also improved the introduction and section 2 of the article. To address this question we have also added a figure in section 2 where we relate AIRS cloud temperature, the IR brightness temperature and the cloud emissivity.
We also have written more details in comparison with other studies in a discussion session at the end of the paper.

**7.** Just because the cloud top temperature is correlated with the rain rate does not mean that it is a good proxy for convective strength or intensity. In fact, a recent study has shown that the highest rain rates in the tropics may not be associated with the most intense convection based on radar reflectivity echo tops (Hamada et al.  2015, Nature Communications, http://www.nature.com/articles/ncomms7213?WT.ec_id=NCOMMS-20150225). Of   course, this   is   using   the   traditional   definition   of   convective   intensity, which   typically   refers   to   the   updraft   vertical   velocity   magnitude. It   has   been   known   for   some   time   though   that   convection   need not be   intense   to   reach   the   tropopause   (e.g.,   Zipser   2003, Meteorological Monographs, http://link.springer.com/chapter/10.1007/978-1-878220-63-9_5). Therefore,  I  recommend removal of the "convective intensity" terminology because it is being very loosely, which leads to confusion.  Why not simply refer to the "convective depth" instead? It could still be pointed out that it is positively correlated with rain rate.

We have added a discussion on the terms convective intensity / strength and depth in the introduction and in the discussion of the results of the relationships, section 4.  Indeed, for our proxy TCbmin it is more appropriate to use the term convective depth, and therefore we have replaced it in the manuscript.

**8.** On page 8,  line 1-2,  it is stated that minimum brightness temperature has been  shown  to  be  a  more  skillful  proxy  to  describe  convective  intensity compared  to  the radar echo height based on Jiang (2012), but this is not what is concluded in Jiang (2012). Jiang (2012) states that minimum infrared brightness temperature in the inner core of tropical cyclones is a better indicator of tropical cyclone rapid intensification than other proxies for convective intensity. It is also problematic that this study, which focuses on tropical cyclones, is being used in the manuscript as representative of all tropical MCSs. In fact, it is well known that for a given minimum infrared brightness temperature, convective

intensity is far stronger over land than ocean, which is reflected in far different reflectivity profiles, microwave brightness temperatures, and lightning flash rates, which are the traditional measures of convective intensity (e.g., Zipser et al. 2006, BAMS, http://journals.ametsoc.org/doi/pdf/10.1175/BAMS-87-8-1057; Liu et al. 2007, J. Climate, http://journals.ametsoc.org/doi/full/10.1175/JCLI4023.1; many others). Minimum infrared brightness temperature is assuredly not the most skillful proxy for convective intensity when used across the entire tropics.

**This has been rectified in the manuscript as written for comment 7.**

**9.** Is Figure 1 just one example of many possibilities or is it an average relationship? Does a pair of given brightness temperature and emissivity values always produce the same retrieved cloud top temperature? For example, will a brightness temperature of 260 K and emissivity of 0.6 always produce a cloud top temperature of 230 K? More information on how cloud top temperature is retrieved and limitations of the retrieval would be helpful.

As written before, we have improved the description of the AIRS cloud retrieval. Figure 1 provides a statistical study; to make it clearer we have A) added quartile bands and **added text** to better explain the distinction between AIRS temperature and TB. B) Added a new 3d plot, Fig. 2 in the manuscript, with Tb and T cloud in x and y axis, respectively, and in the z axis the average emissivity in each (T,TB) bin.

**10.** Possible inconsistencies between Figure 7 and Figures 10 and 12 need to be explained. For example, in Figure 7a, cloud system size increases from stage 6 to stage 9, but in Figure 7e, the minimum cloud top temperature increases from stage 6 to stage 9 (during system maturity when convective area fraction is between 0.1 and 0.3), so as minimum cloud top temperature increases, system size increases. However, Figure 10 shows system size decreasing with increasing minimum cloud top temperature, which is the opposite relationship. Similarly, Figure 12 shows that the thin cirrus anvil area fraction decreases with increasing minimum cloud top temperature, but Figure 7c and 7e show that thin cirrus anvil area fraction increases with increasing minimum cloud top temperature, which is the opposite relationship.

Fig. 9 was used to determine convective systems which are relatively mature, corresponding to a plateau in convective core size, after it has increased during maturing. From Fig. 9b we select cloud systems corresponding to maturity steps 7 -9 (Cb fraction 10 – 30%). For these intervals we have on average similar average cloud system properties. All following analyses in which we explore a relationship between convective depth and anvil properties are done for these mature systems (Figs. 11 – 15), for which the average properties are similar, but for which we see an increase in cloud system size, precipitation and ratio of thin cirrus/total anvil when the convective depth increases.
A better choice would have been probably CB fraction between 15%-25%, but this would shrink the statistics. We have however redone the analyses with this definition of maturity and the results are very similar (see Fig. 6 below).

**11.** Are the results in Figure 14 a result of different life cycle stages of the single core systems or do these differences also exist for a given life cycle stage (indicating differences in the life cycles of systems of varying convective depth)? Clarification here would provide valuable insight into the results.

Fig. 15 was performed for **mature** single core systems. We have clarified this in the manuscript. To assess if the threshold of Cb fraction between 10% and 30% is

not too broad, we have analyzed the relationships separately for 4 smaller intervals of convective fraction inside the maturity range, and the results are very similar for all 4 bins in all 3 in TCbmin, see Fig.6.

[Figure]

**Figure 6.** Emissivity within single core **mature** cloud systems as a function of the normalized distance to the convective core for four subintervals of convective fraction, separated in three groups wrt to convective intensity; Left: systems with $T_{cb}^{min}$ <=200K, middle: 200<$T_{cb}^{min}$ <=230K  and right: $T_{cb}^{min}$ > 230K.

**Minor Comments**
1. I suggest changing "build" on page 1, line 23 to "are part of" since the clouds are primarily a function of the convection rather than the other way around. OK
2. On page 2, line 5, there appears to be a missing word after "MCS's". added word 'anvil'
3. The data is gridded at a resolution of 0.5 , but it seems that the distribution of cloud types defined at the native measurement resolution in each grid box is used for all of the figures. Is this correct? Added a phrase explaining that indeed the information on the individual occurrence of each cloud type in each grid is used to compute cloud type fractions.  However, when it comes to physical properties, the average values over the grid are used.
4. The gap between orbits is largest at the equator.  Please state the width of the gap and scan at the equator on page 4, line 19. Added these numbers in the parenthesis
5. Which latitude band is used for most of the figures?  This should be mentioned in Section 2.Added this information in the abstract, and made it clear in section 2
6. Please capitalize "south" on page 5, line 26. OK
7. Remove "s" from "includes" on page 6, line 9. OK
8. Insert "do" after "dissipation" on page 6, line 24.  OK
9. Change "is" to "are" on page 7, line 18.  OK
10. Change "on" to "in" on page 7, line 22.  OK
11. Remove "or" on page 7, lines 24 and 25.  OK
12. Change "signal" to "reflectivity" and "or brightness temperature" to "and microwave or infrared temperatures" on page 7, line 25.  OK
13. Insert "cloud top" before "temperature" on page 7, line 31.  OK
14. Does the resolution of the minimum retrieved cloud top temperature used in analyses change based on distance from nadir?

pcld and therefore Tcld should not depend on viewing angle, since the clear sky and cloudy radiances used to determine cld emissivity in the chi2 method are simulated for the corresponding viewing angles. To determine the minimum temperature within a convective core we use the
average Tcld per grid so it is a conservative estimation.

15. Please clarify what is meant by "rain detection offset over land" on page 8, line 8.   we removed this phrase
16. Change "is" to "it" on page 8, line 31. OK
17. A citation is needed for the statement that convective intensity will increase in a warming climate on page 8, lines 32-34.  Added reference to: Tan et al. 2015, Bony et al. 2016
18. Insert "of" after "years" on page 9, line 2. OK
19. Why  do  the  distributions  in  Figure  6  go  less  than  0  and  greater than 1  when convective fraction cannot be less than 0 or greater than 1?
 It is a  smoothen kernel density estimate. We have added the histogram beneath so that there is not confusion.

20. Are the bars in Figures 7 and 9-13 standard errors of the mean? If so, please state that and include the sample sizes used to make the figures.

Yes these are standard errors and the statistics for single core systems is the one shown in figure 8 i.e about 130K systems while about 17K systems after the filtering for multi-core systems are used in plots 9 to 13.

[revised manuscript text omitted]

---

## Author Response (AR2)

**ANSWERS TO REFEREE #2**

**Overview**

I thank the authors for carefully considering and thoroughly responding to my comments. They have put in a lot of effort to revise the manuscript, and I am happy with most of their changes. Therefore, I only have some minor comments and suggestions.

**Minor**

Comments

1. I don't question that convective cores contain or once contained active deep convection, so I don't think the addition of the ERA vertical velocity is necessary. I simply wanted a clear statement in the text that these regions often include stratiform precipitation in addition to convective precipitation, which is now included, so including or excluding the new ERA vertical velocity analysis is up to you. If it is kept though, I suggest removing the term "updrafts" though because updrafts typically refer to smaller scales than the ERA analysis. Vertical motion at the ERA resolution can correspond to embedded convective motions or larger-scale ascent associated with waves or stratiform precipitation.

We decided to maintain the vertical wind information, however, as indeed the spatial resolution is low, we replaced the term 'updraft' with 'large-scale ascent'.

2. Is the definition of land and ocean (fraction > 0.5) included anywhere in the text? If not, can this definition be added to the text?

We gave this information by adding (bold) on page 8 "Single core systems over land and ocean, the former having a fraction of land convective grid cells over total convective grid cells above 0.5 and the latter below 0.5, are further separated to early afternoon (PM) and night (AM), since diurnal variations are expected."

3. What is meant by "less convective ocean regions" on page 10, lines 28-29 in the track changed version of the manuscript? Please clarify.

Indeed it is misleading, we have replaced the 'in the less convective ocean regions' with 'in the Atlantic and E. Pacific regions'

Before addressing the remaining questions we kindly remind the reviewer that the proxy for the convective depth is not the **brightness temperature** but the **convective core minimum cloud temperature**, a retrieved quantity which is directly linked to the cloud altitude, as has been indicated in the previous review round, see answers to question 2, 6 and 9 and section 2.1 and 2.2 of the manuscript.

4. 1. I like the addition of regional relationships, but it would be nice to point out that in addition to having similar relationships to all tropical land and ocean regions in general, relationships between different land regions or different ocean regions do differ, especially for minimum brightness temperature and thin cirrus/anvil fraction, while rain rates are quite variable between different land regions. It also seems that convective intensity related metrics (minimum Tb, rain rate) differ the most early in the life cycle, while area related metrics differ the most late in the life cycle. Again, it is up to you whether to add information, but I find it to be relevant to interpreting the larger-scale tropical land and ocean results.

Indeed, even though the physical properties of the cloud systems follow the same behavior (except the  $T_{cb}^{min}$ ) as a function of the convective core fraction for all

regions, the absolute values may differ between regions. In the end of section 3.2 we have therefore added a paragraph pointing out these differences.

5. The in depth first paragraph added in Section 4 discussing results in Takahashi and Luo (2014) seems out of place in a results section, since it doesn't discuss results from this manuscript, and I'm confused as to its relevance to interpreting the results. I suggest cutting this down or removing it. Additionally, the second to last paragraph in Section 4 seems like a rather large aside, again getting into convective intensity and relationships with environmental conditions and radar measurements, which again, seems off topic, especially since the discussion of the results is in the context of convective depth rather than intensity. Not that intensity can't be mentioned, but focusing on it as much as it now is in these paragraphs draws attention away from the main points of the paper.

We have trimmed the first paragraph of section 4 accordingly (page 10).

6. Although an increase in the size of systems is observed with increased convective depth in different regions, it may be important to note that the slope of this relationship changes, especially for ocean regions, where the W. Pacific is similar to land regions rather than the other ocean regions (E. Pacific and Atlantic), so it would seem that the ocean points in Figure 11 are not representative of all ocean regions, and therefore, the statement on page 12, lines 31-33 in the track changed version isn't true for the W. Pacific.

We have added some text mentioning these regional differences, and also added a phrase in the conclusion section mentioning the regional analysis cross check.

7. Similarly, for thin cirrus/anvil fraction as a function of minimum Tb, I agree that the general relationship of the fraction increasing with decreasing minimum Tb occurs in all regions, but the regional spread of the magnitude of the thin cirrus/anvil fraction for a given minimum Tb is large, which I think should be noted in the text.

We have added text accordingly.

[revised manuscript text omitted]